# The Polycomb protein Ezl1 mediates H3K9 and H3K27 methylation to repress transposable elements in *Paramecium*

Andrea Frapporti [1,6,7], Caridad Miró Pina [1,7], Olivier Arnaiz [2], Daniel Holoch [3], Takayuki Kawaguchi [1], Adeline Humbert[1], Evangelia Eleftheriou[2], Bérangère Lombard[4], Damarys Loew [4], Linda Sperling [2], Karine Guitot [5], Raphaël Margueron[3] & Sandra Duharcourt [1]

In animals and plants, the H3K9me3 and H3K27me3 chromatin silencing marks are deposited by different protein machineries. H3K9me3 is catalyzed by the SET-domain SU(VAR)3–9 enzymes, while H3K27me3 is catalyzed by the SET-domain Enhancer-of-zeste enzymes, which are the catalytic subunits of Polycomb Repressive Complex 2 (PRC2). Here, we show that the Enhancer-of-zeste-like protein Ezl1 from the unicellular eukaryote *Paramecium tetraurelia*, which exhibits significant sequence and structural similarities with human EZH2, catalyzes methylation of histone H3 in vitro and in vivo with an apparent specificity toward K9 and K27. We find that H3K9me3 and H3K27me3 co-occur at multiple families of transposable elements in an Ezl1-dependent manner. We demonstrate that loss of these histone marks results in global transcriptional hyperactivation of transposable elements with modest effects on protein-coding gene expression. Our study suggests that although often considered functionally distinct, H3K9me3 and H3K27me3 may share a common evolutionary history as well as a common ancestral role in silencing transposable elements.

[1] Institut Jacques Monod, Université de Paris, CNRS, 75013 Paris, France. [2] Institute for Integrative Biology of the Cell (I2BC), CNRS, CEA, Univ. Paris-Sud, Université Paris-Saclay, 91198 Gif-sur-Yvette CEDEX, France. [3] Institut Curie, Paris Sciences et Lettres Research University, INSERM, U934, CNRS, UMR3215, Paris 75005, France. [4] Institut Curie, Paris Sciences et Lettres Research University, Centre de Recherche, Laboratoire de Spectrométrie de Masse Protéomique, 26 rue d'Ulm, 75248 Cedex 05 Paris, France. [5] Sorbonne Université, Ecole Normale Supérieure, Paris Sciences et Lettres Research University, CNRS, Laboratoire des biomolécules, LBM, 75005 Paris, France. [6] Present address: The Gurdon Institute, University of Cambridge, Cambridge CB21QN, UK. [7] These authors contributed equally: Andrea Frapporti, Caridad Miró Pina. Correspondence and requests for materials should be addressed to S.D. (email: sandra.duharcourt@ijm.fr)

Post translational modifications of histones play essential roles in DNA transactions at the level of chromatin. Among these modifications, trimethylation of histone H3 on lysine 9 (H3K9me3) and lysine 27 (H3K27me3) are typically associated with transcriptionally silent chromatin. These two epigenetic signatures are controlled by distinct families of histone methyltransferases. H3K9me3 is catalyzed by the SET-domain SU(VAR) 3–9 enzymes[1–5], while H3K27me3 is installed by the SET-domain Enhancer-of-zeste enzymes, acting as the catalytic subunits of Polycomb Repressive Complex 2 (PRC2)[6–9].

It is widely believed that H3K9me3 and H3K27me3 do not co-occur at the same genomic loci, and this assumption is supported by evidence from several organisms. In human and fly, the majority of H3K9me3-enriched domains are concentrated at repeated DNA sequences, such as transposable elements and satellites, and repress the transcription of the underlying repeats[10–13]. In contrast, H3K27me3 is involved in the stable and heritable maintenance of cell-type-specific gene repression[14].

However, this view has come to be challenged by examples of co-occurrence of H3K9me3 and H3K27me3. Indeed, ChIP-seq data indicate that these two marks can be found together at a subset of developmentally regulated genes in mouse embryonic stem cells, extra-embryonic lineages and human differentiated cells[15–17]. In addition, mass spectrometry experiments have demonstrated that H3K27me3 can coexist with H3K9me3 on the same histone octamer in embryonic stem cells[18].

An increasing body of evidence suggests that the two types of silent chromatin can compensate for each other, and that H3K9me3 usually prevents the installation of H3K27me3. In mouse embryonic stem cells, the loss of H3K9me3 at pericentromeric regions that occurs upon knockout of SUV39H1 enzymes is accompanied by ectopic establishment of H3K27me3[13]. Moreover, in several human cancer cell lines, pericentromeric regions show massive recruitment of Polycomb Group proteins[19,20]. H3K27me3 has also been shown to relocalize to and silence transposons in mouse embryonic stem cells upon loss of DNA methylation[21,22]. Whether H3K27 and H3K9 trimethylation exert overlapping roles under physiological circumstances remains an open question.

The unicellular eukaryote *Paramecium tetraurelia* shares the highly conserved trimethylation of K9 and K27 on histone H3. In this organism, two distinct types of nuclei with different functions and genome architectures coexist in the same cytoplasm. The diploid micronuclei (MIC, 2n) contain the germline genome that is transmitted to sexual progeny after meiosis. A transcriptionally active somatic macronucleus (MAC, 800n) contains a reduced genome streamlined for gene expression. At each sexual cycle, the parental MAC is lost, while new MICs and MACs, destined for the progeny, develop from a copy of the diploid zygotic nucleus. In the new developing MAC, the germline genome is endor-eplicated to reach its final ploidy of ~800n and undergoes massive programmed DNA elimination (for review[23]). At least 25% of the ~100 Mb MIC genome is removed, including all repeated sequences (transposable elements, tandem repeats…)[24].

We have previously shown that the Enhancer-of-zeste-like protein Ezl1 is essential for programmed DNA elimination and viability of the sexual progeny[25]. Here, we provide evidence that, *Paramecium* Ezl1, which exhibits significant sequence and structural similarities with human EZH2, displays distinct enzymatic properties. The Ezl1 methyltransferase has a broader substrate specificity, including histone H3K27 and K9, and potentially secondary residues in vitro. Consistent with these observations, a catalytic mutant of Ezl1 allows us to induce extensive and simultaneous loss of H3K9me3 and H3K27me3 in vivo. By combining transcriptional and chromatin profiling, we find that the H3K27me3 and H3K9me3 marks co-occur at

multiple families of transposable elements repressed by Ezl1. Our work suggests that the ancient family of Enhancer-of-zeste proteins may initially have evolved to silence mobile genetic elements.

## Results

**Comparative homology modeling of the catalytic SET domain.** The SET domain of the *Paramecium* Ezl1 protein, ~130 amino acids in length, shows 33% sequence identity with that of human EZH2, the PRC2 catalytic subunit (Fig. 1a). Based on the recently published crystal structure of human EZH2 in complex with a histone H3K27M peptide and S-Adenosyl-L-homocysteine (SAH)[26], we performed homology modeling of the SET domain of Ezl1 (Supplementary Fig. 1). It exhibits a pseudoknot-like structure characteristic of the catalytic part of histone lysine methyltransferases[27]. The methyl group from the cofactor S-adenosylmethionine (SAM) is bound on one side of the pseudoknot, while the ε-amino group of the lysine residue to which it is transferred is on the opposite side (Supplementary Fig. 1).

EZH2 and the modeled Ezl1 are highly conserved in structure. The aromatic residues forming the lysine binding channel are conserved in Ezl1 (Fig. 1b). The SAM binding pocket residues are conserved or substituted either with different amino acids that do not alter the hydrogen bond network between the amide backbone and the cofactor, or with amino acids whose side chains have similar properties (Fig. 1c). Interestingly, phenylalanine Phe-665 is substituted with a tyrosine residue (Tyr-502) in Ezl1, likely providing an additional contact with the cofactor, as is the case for methyltransferases that catalyze H3K9 methylation (Fig. 1c)[28].

A feature of the histone H3 substrate-binding site of human EZH2 is the well-defined pocket accommodating the Arg-26 residue in the (−1) position. This residue makes a series of hydrogen bonds to the complementary side chains of the I-SET domain and is crucial for the recognition of the target lysine. Even though the helix structure of the I-SET domain is maintained, the amino acid side chain distribution is rather different in the modeled Ezl1 structure. Indeed, polar residues (Asp-652 and Asp-659) in EZH2 are substituted with two apolar residues (Met-489 and Trp-496) in Ezl1 (Fig. 1d). Consequently, the substrate H3K27M is anchored in a different way. Only a few of the described contacts with the H3K27M peptide are strictly maintained in the modeled Ezl1 structure (Gln-648 (Asp-485 in Ezl1), Leu-666, Asn-668 (Glu-505 in Ezl1), Ala-697, Tyr-726). We hypothesize that additional interactions with the Asp-485, Ser-488, Glu-492, and Glu-505 residues stabilize the arginine residue in the (−1) position (Fig. 1d). Interestingly, the position corresponding to Glu-505 contains a residue with similar properties in the GLP and G9a histone K9 methyltransferases (Asp-1145 and Asp-1088, respectively). Indeed, this aspartic acid residue makes electrostatic interactions with the Arg-8 residue, in the (−1) position of the histone K9 peptide[26]. In the modeled Ezl1 structure, the hydrogen bond network appears to be compatible with anchoring of an arginine residue at the (−1) position for recognition of either lysine 27 or lysine 9 of the histone H3 substrate.

Sequence comparison reveals a substitution of an alanine (A677) in EZH2 with a glycine (G514) in Ezl1 (Fig. 1a). A glycine at the equivalent position is also observed for the SET domain-containing Dim-5 from *Neurospora crassa* (Fig. 1e), which has been reported to trimethylate its H3K9 substrate[29]. Structural comparison of the active sites of Ezl1 and Dim-5 shows that residues surrounding the lysine channel are highly similar in Ezl1 and Dim-5 (Fig. 1e). The A677G substitution has been described as a gain-of-function mutation in EZH2, found in lymphoma cell

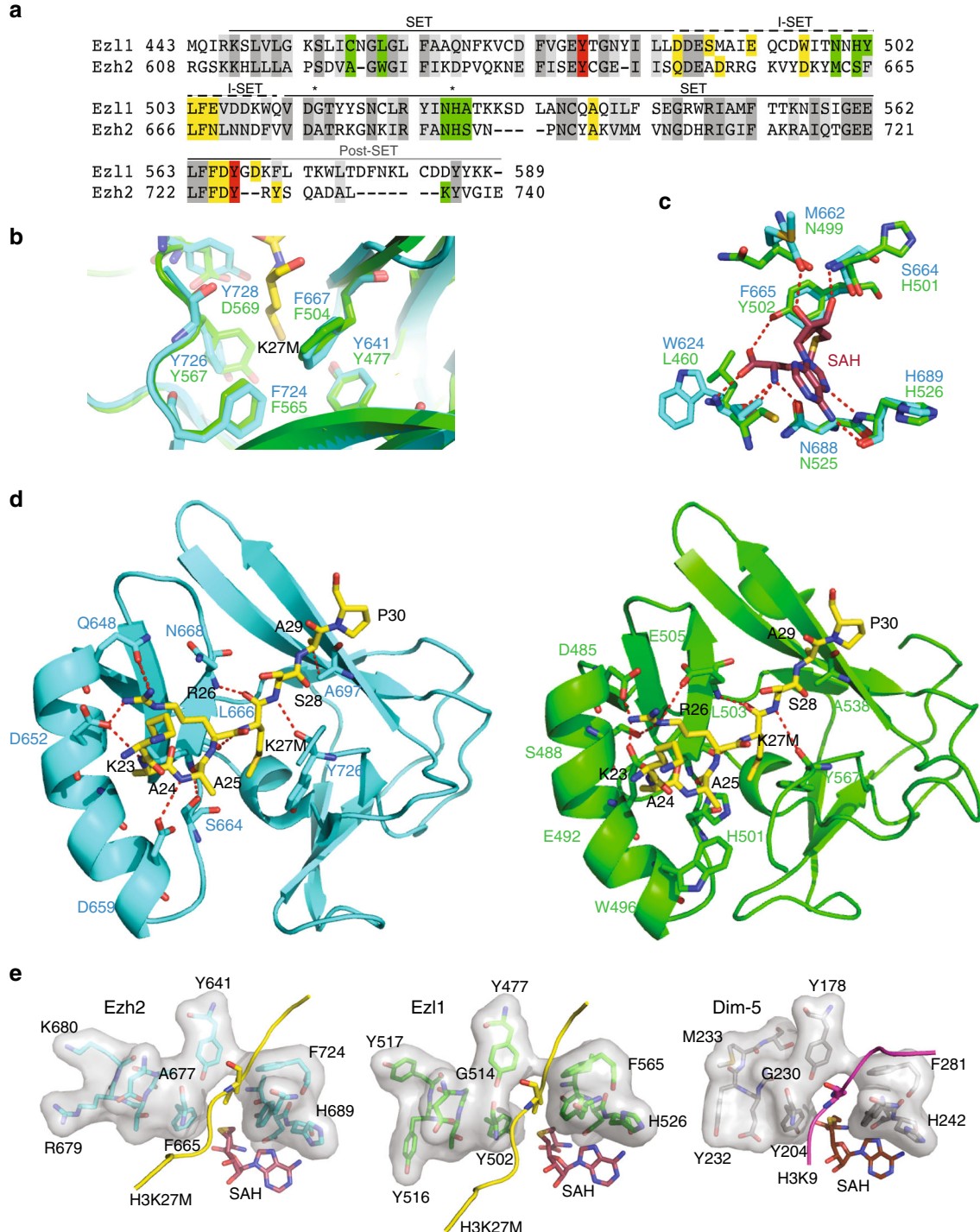

**Fig. 1** Comparative sequence and homology modeling of the catalytic SET domain of *Paramecium* Ezl1 and human EZH2. **a** Sequence alignment of the SET domains of *Paramecium tetraurelia* Ezl1 (PTET.51.1.G1740049) and human EZH2 (612–727; Q15910). The I-SET domain (648–674 in EZH2) is indicated. Substrate and cofactor-binding residues as well as important residues for catalysis are highlighted in yellow, green, and red, respectively. Identical residues are colored in gray and positively substituted residues in light gray. Positions A677 in EZH2 and H526 in Ezl1 are indicated by asterisks. **b** Conservation of catalytic residues. Structural alignment of human EZH2 SET domain (cyan) (PDB ID: 5HYN) bound to the peptide inhibitor H3K27M (yellow) with modeled Ezl1 (green). Superposition of the lysine binding channel shows the conserved aromatic residues: F724/565, Y726/567, F667/504, Y641/477 in EZH2 and Ezl1, respectively. Note that tyrosine residue (Tyr-728) in EZH2 is replaced by an aspartic residue (Asp-569) in Ezl1, likely because of misalignment in the post-SET domain. **c** Conservation of SAM-binding residues. Structural alignment of human EZH2 SET domain (cyan) bound to SAH cofactor (red) with modeled Ezl1 (green). Hydrogen bonds are represented by red dashed lines. **d** Structural comparison of the SET domain of human EZH2 (cyan) bound to H3K27M peptide (yellow) (left panel) with modeled Ezl1 (right panel). Substrate interacting residues are represented as sticks. Hydrogen bonds are represented by red dashed lines. **e** Surface representation of the catalytic pockets of human EZH2 with H3K27M peptide (yellow) and SAH (red) (PDB ID: 5HYN), modeled Ezl1 and *Neurospora crassa* Dim-5 with H3K9 peptide (magenta) and SAH (red) (PDB ID: 1PEG)

lines and primary tumors[30]. Interestingly, the modeled structure predicts a larger catalytic pocket in Ezl1 than in EZH2 (Fig. 1e), consistent with the increase of the dimensions of the lysine tunnel reported for the EZH2 A677G mutant[30]. As observed for the EZH2 mutant, it might confer a strong enzymatic activity to Ezl1 toward H3K27.

Thus, sequence and structural comparison of Ezl1 SET domain with that of other histone methyltransferases reveals strong similarities with human EZH2, consistent with previous phylogenetic analyses[25]. The most significant differences occur in the I-SET region that contributes to the substrate-binding site, suggesting that Ezl1 may have different substrate preferences from EZH2.

**Ezl1 targets methylation of both K9 and K27 of histone H3**. To investigate the enzymatic activity of the Ezl1 protein, we obtained *Paramecium* transformants expressing full-length, GFP-tagged Ezl1, either wild-type or carrying a histidine-to-alanine substitution (H526A) in the highly conserved RXXNH cofactor-binding motif within the SET domain (Fig. 1a, c), expected to abolish catalytic activity[3,31,32]. Endogenous Ezl1 protein expression was specifically depleted through RNA interference while the wild-type and mutant transgenes were designed to be RNAi-resistant (see Methods). Both GFP fusion proteins were detected at the expected molecular weight, ~97 kDa (Supplementary Fig. 2). We enriched the GFP fusion proteins from nuclear extracts isolated during MAC development by immunopurification (GFP-Trap IP) (Supplementary Fig. 2). Enzymatic activity of the IP was then assessed by monitoring the transfer of a radiolabeled methyl group to recombinant histones (Fig. 2a). Reconstituted mammalian PRC2 core complex was used in parallel as a positive control. We found that, when incubated with histone octamers or oligonucleosomes, GFP-Ezl1 immunoprecipitates transferred the labeled methyl group exclusively onto histone H3 (Fig. 2a, lanes 3–5, Fig. 2b). PRC2 exhibited a similar selectivity toward histone H3, as expected (Fig. 2a, lane 2). In contrast, sharply reduced signals were detected with extracts from the Ezl1[H526A] mutant (Fig. 2a, lanes 6–8), while no signal could be detected in the mock control from non-transformed cells (Supplementary Fig. 2). We conclude that the Ezl1 protein is capable of catalyzing de novo methylation of histone H3 through its catalytic SET domain.

Purified PRC2 is known to methylate histone H3 on K27 and K9 in vitro, even though the activity towards K9 is low[7–9]. To address the lysine specificity of the Ezl1 protein, we performed in vitro methylation reactions on recombinant histone octamers bearing lysine-to-alanine mutations on histone H3 either at K9 (K9A) or K27 (K27A), or both mutations on the same histone H3 tail (K9A/K27A). We used *Xenopus* histones, despite some sequence divergence with *Paramecium* histones (Supplementary Fig. 2), in order to directly compare Ezl1 activity with that of PRC2. Reconstituted mammalian PRC2 core complex showed a very strong preference for K27 as a methylation substrate, with only a weak activity directed towards K9 (Fig. 2c, lanes 1–4), as previously described[7–9]. In contrast, the methyltransferase activity of GFP-Trap Ezl1 immunoprecipitates was moderately reduced on both K9A and K27A single mutants, compared to wild-type octamers (Fig. 2c, lanes 6–8). A stronger reduction was observed on double K9A/K27A mutant octamers (Fig. 2c, lane 9), approaching the low level observed for the GFP-Ezl1[H526A] mutant on wild-type octamers (Fig. 2c, lane 10). The striking difference in activity on the K27A and K9A/K27A mutant octamers, which was not observed for PRC2, strongly suggests that H3K9 is one of the principal substrates of Ezl1.

The 15% residual activity detected with the double K9A/K27A H3 mutant may represent a contaminating activity in our

preparations or, alternatively, a weak activity of Ezl1 toward other residues of H3. To distinguish between these possibilities, we purified Ezl1 from soluble nuclear protein extracts using a stringent FLAG-HA double-affinity immunopurification (see Methods). In vitro methyltransferase activity was strongly reduced on K9A/K27A double-mutant H3, confirming that lysines 9 and 27 are the main targets of Ezl1 (Supplementary Fig. 2). We then performed in vitro methylation reactions with H3-H4 tetramers containing recombinant *Paramecium* H3, instead of *Xenopus* H3, either wild-type or bearing lysine-to-arginine mutations (K9R and K27R single mutants, and K9R/K27R double mutant). Arginine substitution was chosen to maintain the positive charge of the residues. Recapitulating our above results, reduced methyltransferase activity was detected on K9R/K27R double-mutant H3 compared to wild-type H3 or single mutants (K9R and K27R) (Fig. 2d and Supplementary Fig. 2), indicating that Ezl1 also targets both lysines 9 and 27 in its conspecific H3 substrate. The H526A mutation severely compromised the methyltransferase activity of Ezl1 in this setting as well (Fig. 2d).

To investigate which lysine residue of *Paramecium* H3 is methylated in an Ezl1-dependent manner, we performed quantitative label-free mass spectrometry analysis. In vitro methylation reactions were performed with tetramers containing *Paramecium* histone H3 and with FLAG-HA double-affinity purified Ezl1[wt] and Ezl1[H526A], or with tetramers only as negative controls. Under the assay conditions, Ezl1 catalyzes trimethylation of K9 and K27 (Fig. 2e, Table 1, and Supplementary Data 1). We could detect the existence of other sites of methylation within the N-terminal part of histone H3, in particular K23 (Supplementary Data 1), but only the methylations on K9 and K27 are significantly reduced in the catalytic Ezl1[H526A] mutant (Table 1, Fig. 2e, and Supplementary Data 1).

Detection of methylation on other lysines provides an explanation for the residual activity we detected when lysines 9 and 27 were mutated, either on *Xenopus* or *Paramecium* H3 (Fig. 2c, d and Supplementary Fig. 2). Using quantitative label-free mass spectrometry analysis with K9R/K27R double-mutant *Paramecium* H3, we indeed found that K23 is methylated in an Ezl1-dependent manner when lysines 9 and 27 are no longer substrates for methylation by the enzyme (Supplementary Data 1). Altogether, our data support the hypothesis that lysines 9 and 27 are both major substrates of Ezl1, while K23 is a secondary site of methylation whose in vivo relevance would require further investigation.

**Role of Ezl1 catalytic activity in vivo**. To address the role of Ezl1 catalytic activity in vivo, we performed a genetic complementation assay by expressing the wild-type or mutant *GFP-EZL1* transgenes in cells in which the endogenous *EZL1* gene was knocked down. The GFP fusion proteins were localized in the zygotic MAC of the transformed cells in both control and *EZL1* RNAi conditions, indicating that each RNAi-resistant transgene produces a fusion protein with correct nuclear localization (Fig. 3a, b)[25]. No toxicity was observed during vegetative growth or following sexual events of transformed cells grown under control conditions, indicating that the wild-type and mutant GFP-*EZL1*[H526A] fusions have no dominant-negative effect.

Knockdown of the *EZL1* gene results in loss of H3K27me3 and H3K9me3 during development, blockage of programmed DNA elimination and lethality of the sexual progeny[25]. To check whether the wild-type and mutant transgenes are functional, transformed cells were subjected to RNAi targeting the endogenous *EZL1* sequence during the self-fertilization sexual process of autogamy (Fig. 3a). During development, H3K9me3

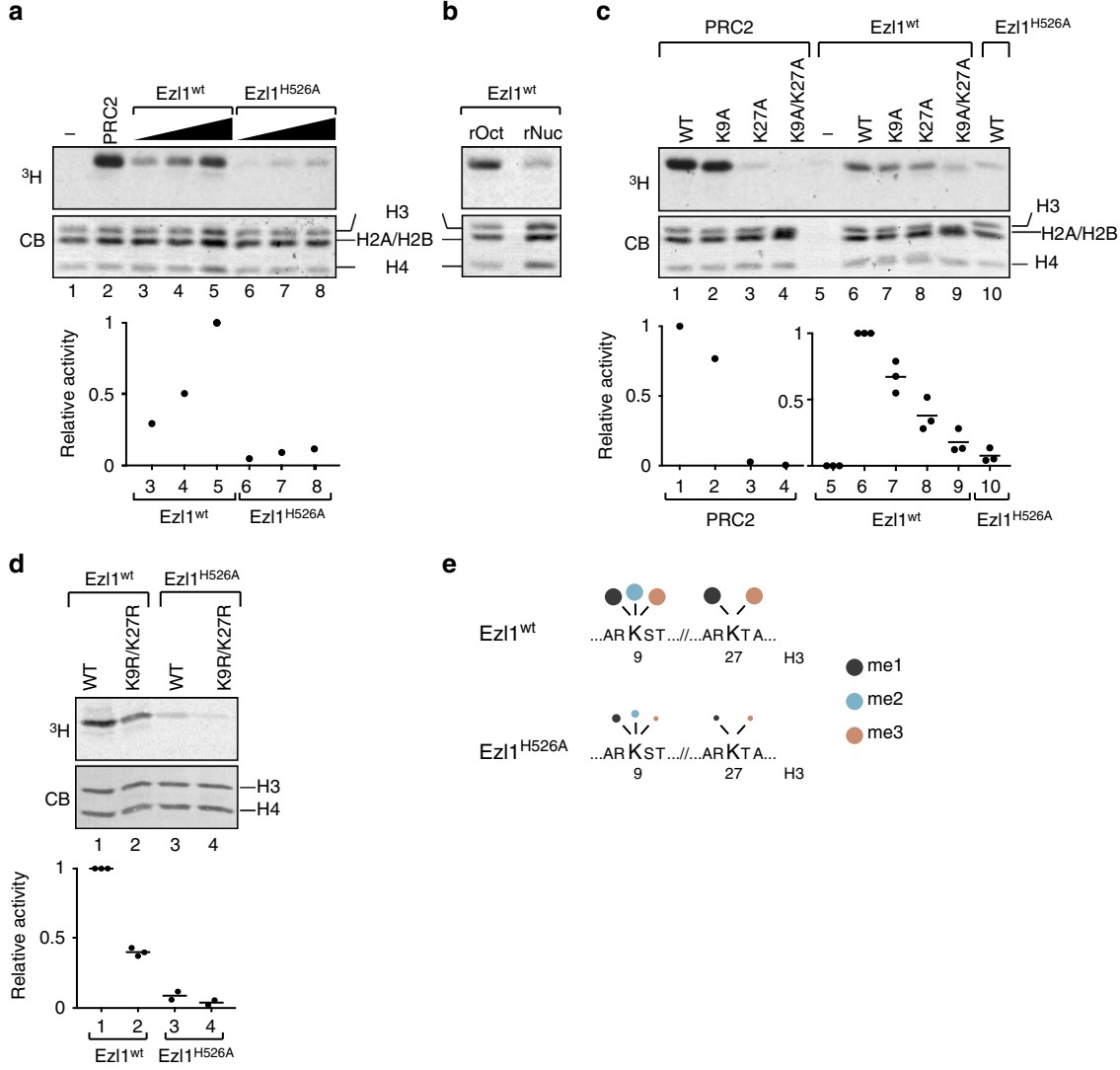

**Fig. 2** Histone methyltransferase assay. Full-length wild-type GFP-Ezl1[wt] and mutant GFP-Ezl1[H526A] proteins were immunoprecipitated at 12 h after the onset of sexual events (see Supplementary Fig. 2) from *Paramecium* transformed cells, in which endogenous Ezl1 protein expression was specifically depleted through RNA interference, and used for in vitro histone methyltransferase reactions (**a–c**). **a** Wild-type *Xenopus* recombinant histone octamers are used as substrates and S-adenosyl-[methyl 3H]-methionine as methyl donor. Affinity-purified recombinant PRC2 complex is used as a positive control. The same reactions with histone octamers only were performed as a negative control. Coomassie stain (CB, bottom panel) shows histones and the autoradiograph ([3H], top panel) indicates H3 histone methyltransferase activity. The graphs show relative quantification of histone methyltransferase signals analyzed by scintillation counting after SDS-PAGE. **b** Histone methyltransferase activity on recombinant histone octamers (rOct) and on oligonucleosomes (rNuc). **c** Substrate specificity. Histone methyltransferase assays were performed with recombinant histone octamers assembled with wild-type H3 histones or with H3 histones bearing a point mutation on lysine 9 (K9A) or on lysine 27 (K27A) or on both (K9A/K27A) on the same histone H3 tail. The same reaction without addition of histone octamers was performed as a negative control. For Ezl1[wt] and Ezl1[H526A], the horizontal bars indicate the mean of three biological replicates (independent nuclear extracts). Circles indicate the individual data points. **d** Full-length 3xFLAG-HA tagged Ezl1[wt] and Ezl1[H526A] proteins were immunoprecipitated using a FLAG-HA double-affinity purification from *Paramecium* transformed cells, depleted for the endogenous Ezl1 protein in the case of the mutant FLAG-HA-Ezl1[H526A], and used in in vitro histone methyltransferase reactions (see Supplementary Fig. 2). Histone methyltransferase assays were performed with recombinant histone tetramers assembled with wild-type H3 from *Paramecium* or with *Paramecium* H3 histones bearing point mutations on both K9 and K27 (K9R/K27R) on the same histone tail. The horizontal bars indicate the mean of two or three biological replicates (independent nuclear extracts) for Ezl1[wt] and Ezl1[H526A], respectively. **e** Schematic representation of quantitative label-free mass spectrometry analysis of *Paramecium* histone H3. See Table 1 and Supplementary Data 1. Owing to the insertion of one residue at position 14 in the *Paramecium* H3 sequence (see Supplementary Fig. 2), positions in the *Xenopus* and *Paramecium* H3 sequences differs by one amino acid gap after that position (Supplementary Fig. 2 and Supplementary Data 1). For simplicity, K24 and K28 in the *Paramecium* H3 sequence are referred to as K23 and K27 in the schema and in the manuscript. Source data are provided as a Source Data file

and H3K27me3 deposition were assayed by immunostaining with anti-H3K9me3 and anti-H3K27me3 antibodies, respectively. Recovery of viable sexual progeny after sexual events was measured by transferring individual autogamous cells to fresh growth medium and letting them resume vegetative growth. In

non-injected cells, no H3K9me3 or H3K27me3 could be detected in the developing MACs, and no viable progeny were obtained upon *EZL1* RNAi, indicating that the endogenous *EZL1* gene is efficiently silenced (Fig. 3b, c). Injection of the wild-type *GFP-EZL1* transgene restored both H3K9me3 and H3K27me3

**Table 1 Quantitative label-free mass spectrometry analysis of *Paramecium* histone H3**

| Peptide sequence | PTM position | Sequence in sup.data 1 | Ezl1$^{H526A}$/ Ezl1$^{wt}$ ratio | *p*-value |
|---|---|---|---|---|
| **K**STAGNKKPTK | Trimethyl (K9) | 4 | 0.31 | 0.02 |
| **K**STAGNKKPTK | Methyl (K9) | 9 | 0.27 | 0.01 |
| **K**STAGNKKPTKHLATKAAR | Methyl (K9) | 12 | 0.72 | 0.03 |
| **K**STAGNKKPTKHLATKAAR | Dimethyl (K9) | 14 | 0.46 | 0.03 |
| **K**STAGNKKPTKHLATKAAR | Trimethyl (K9) | 15 | 0.26 | 0 |
| **K**STAGNKKPTKHLAT**K**AAR | Trimethyl (K9) and Methyl (K24) | 16 | 0.3 | 0 |
| **K**TAPAVGATGGLKKPHK | Methyl (K28) | 28 | 0.33 | 0.03 |
| **K**TAPAVGATGGLKKPHK | Trimethyl (K28) | 30 | 0.36 | 0.01 |
| **K**TAPAVGATGGLKKPHK | Trimethyl (K28) | 32 | 0.29 | 0.04 |

In vitro methylation reactions were performed with FLAG-HA double-affinity purified Ezl1$^{wt}$ and Ezl1$^{H526A}$, or with tetramers only as negative controls, in three biological replicates. Histone H3 A [Propionyl (N-term)]RTK[Propionyl (K)]QTAR peptide was used as internal reference (see Supplementary Data 1). The modified residues (in bold) as well as the post translation modification (PTM) positions are indicated. Only significantly modified peptides are displayed (*p*-value < 0.05)

deposition and led to recovery of 95% viable progeny (Fig. 3b, c), indicating that the *GFP-EZL1* transgene fully complements the *EZL1* knockdown, and is therefore functional. In contrast, the *GFP-EZL1*$^{H526A}$ mutant transgene was unable to complement an *EZL1* knockdown, as indicated by the complete absence of H3K9me3 and H3K27me3 deposition and the absence of sexual progeny bearing a functional zygotic MAC (Fig. 3b, c).

To analyze the effects of the transgenes on DNA elimination, total genomic DNA was extracted at the end of MAC development, when DNA elimination is normally complete, and used to monitor a few representative loci by quantitative PCR (Fig. 3d). As expected, all DNA elements tested were excised from the new developing MACs in control RNAi, while the same DNA elements were retained upon *EZL1* RNAi in untransformed clones (Fig. 3d). In the presence of the *GFP-EZL1* complementing transgene, DNA elimination levels returned to normal (Fig. 3d). In contrast, expression of the *GFP-EZL1*$^{H526A}$ mutant transgene failed to restore correct DNA elimination (Fig. 3d). Altogether, these results show that the catalytic activity of Ezl1 is required in vivo for the deposition of H3K9me3 and H3K27me3, for the production of viable sexual progeny and for DNA elimination.

**Transcriptional derepression of transposable elements**. To determine whether simultaneous loss of H3K9me3 and H3K27me3 affects transcription, we performed paired-end RNA-seq at different time-points during the life-cycle (Supplementary Table 1). RNA samples were isolated during vegetative growth (V) and during post-zygotic development (T0; T5; T10; T20; T35; T50) of control and *EZL1* RNAi cells. Comparison of steady-state levels of poly-A RNAs between control and *EZL1* RNAi was done by mapping RNA-seq reads onto MAC and MIC reference genomes and counting the fragments that map onto genes, transposable elements (TE) and tandem repeats. Since cells enter meiosis from a fixed point of the vegetative cell cycle[33], which is not synchronized in our cell cultures, there is an asynchrony of at least 5 h in the time point samples. We therefore grouped samples, after normalization of the counts, using principal component analysis (PCA), and treated the samples within groups (Vegetative, Early, and Late; see Supplementary Fig. 3) as biological replicates for differential gene expression analysis (see Methods).

The *P. tetraurelia* genome encodes 41,533 genes (of which 40,640 are protein-coding genes)[34], which account for about 56 Mb of sequence complexity (80% of the MAC genome is coding). The germline MIC genome contains repeated sequences not present in the MAC genome. Since our objective was to detect transcripts both from genes and from repeated sequences, we used a dual mapping strategy. To optimize specificity for gene

mapping, only uniquely mapping reads were considered. To avoid underestimating expression of repeated elements, multiple hits were allowed for mapping to repeated elements. These consisted of tandem repeats ($n = 1020$, 0.43% of the MIC assembly) and TE copies ($n = 3129$, 2.4% of the MIC assembly). The TE copies were identified using a library of 61 manually curated TE consensus sequences classified as TIR, LINE, Solo-ORF, and SINE elements[24]. No autonomous transposons have been identified in the genome and these TE copies mainly seem to be highly degenerate.

To evaluate the differential expression of all genes, TEs and tandem repeats, we first compared mean normalized fragment counts across the life-cycle time course, for control and *EZL1* RNAi samples. The results for the 45,682 genomic elements (41,533 genes, 3129 TE copies, 1020 tandem repeats; see Methods for details of the genome annotation) are presented in Fig. 4a. Strikingly, TEs are almost exclusively expressed in *EZL1* RNAi samples and not in the control. In contrast, the expression of tandem repeats and of genes is similar in EZL1 and control samples. While 0.15% ($N = 63$) of the genes were differentially expressed at the early time point, 5.8% ($N = 2409$) were either up- or downregulated at the late time point (Supplementary Fig. 4 and Supplementary Data 2). Interestingly, differentially expressed genes are enriched in genes known to be developmentally regulated (Supplementary Table 2)[34]. Differentially expressed genes that are upregulated in *EZL1* knockdown correspond to genes which, in control conditions, are expressed at the early time point and repressed at the late time point (Supplementary Fig. 4). This suggests that the Ezl1 protein functions in maintaining transcriptional repression of developmental genes.

Closer inspection of TE copies shows that about half of all the currently annotated TE copies in the genome become expressed during zygotic MAC development upon *EZL1* RNAi. The effect increases during the time course and is most pronounced late in development (Fig. 4b). As shown in the volcano plot representation (Fig. 4c), 739 TE copies (24%) are differentially expressed ($p < 0.05$, fold-change > 2). Strikingly, all the differentially expressed TE copies undergo significant upregulation at the late time point, ranging from 2 to > 500-fold-change in expression, with no case of downregulation (Fig. 4c), a more dramatic effect than seen for the upregulated differentially expressed genes (Supplementary Table 3). While these 739 TE copies belong to all four of the major distinct TE families that have been found in the genome, LINE copies appear to be over-represented (Fig. 4d left panel).

To examine whether the burst of transcription observed from bulk RNA profiling reflected a general trend within each TE family, we measured the transcriptional output of individual copies for each family. We note that each family of TEs includes copies that are de-repressed by loss of H3K9me3 and H3K27me3,

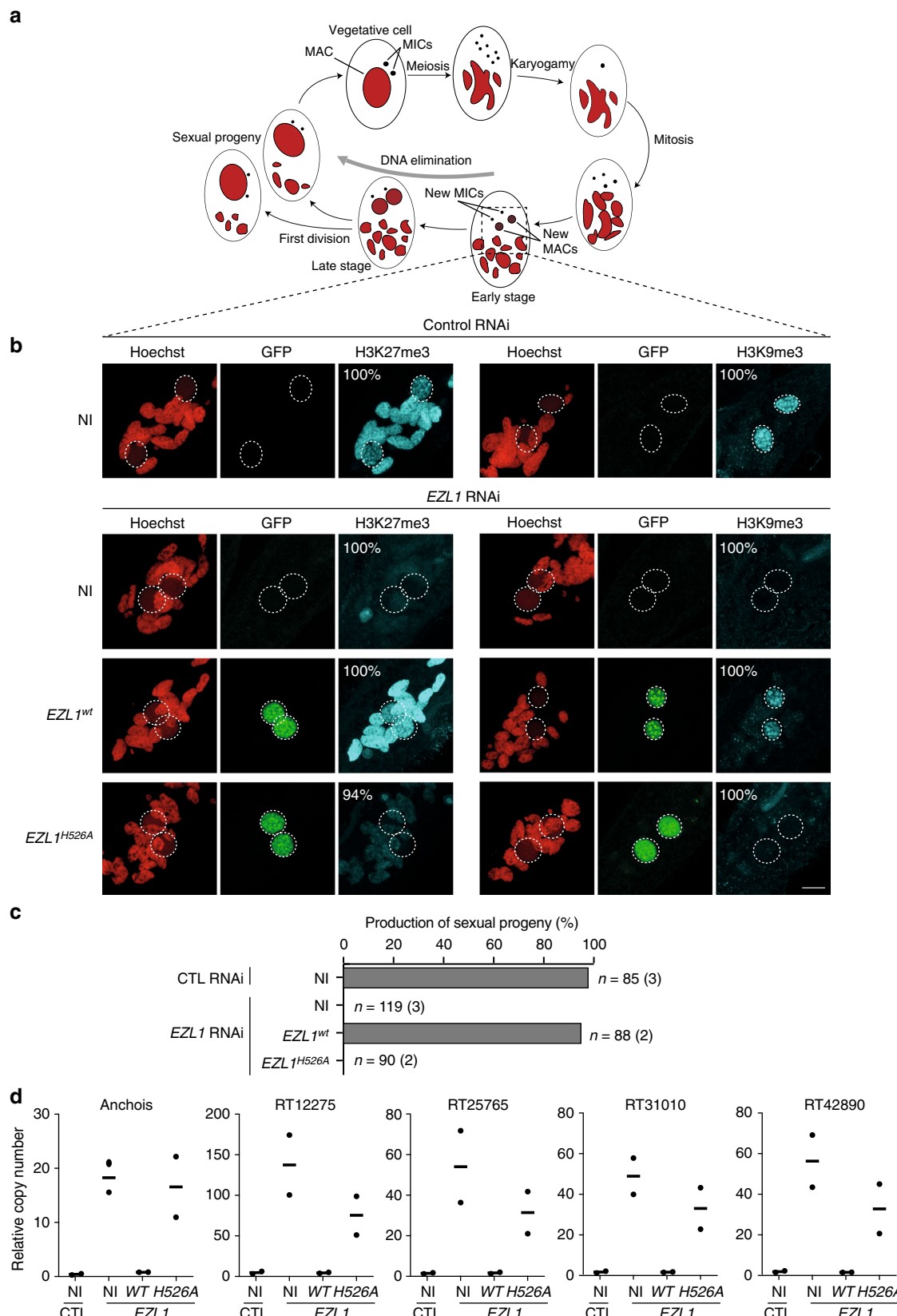

as well as copies that are not affected (Fig. 4d right panel). To examine whether this difference in silencing release correlates with the time since the element was inserted into the genome, we measured the identity of individual copies to the consensus sequence as a proxy for their age. No correlation could be observed between the putative age of the copy and its expression level upon *EZL1* RNAi (Fig. 4e, for LINE copies).

We wanted to rule out that the transcriptional upregulation we observed did not simply reflect the retention of the transposable elements in the zygotic MAC genome upon *EZL1* RNAi. To this

**Fig. 3** In vivo genetic complementation assay. **a** Life-cycle of *Paramecium*. Each cell contains a macronucleus (MAC) and two micronuclei (MICs), both of which divide and segregate to daughter cells during vegetative growth. Starvation triggers MIC meiosis, and one of the meiotic products divides mitotically to form two haploid nuclei. During autogamy, a self-fertilization process, the two haploid nuclei will fuse to produce the zygotic nucleus (karyogamy), which then divides twice to form two new MACs and two MICs. During development of the new MAC, the MIC genome undergoes massive programmed DNA elimination. Upon nutrient supply, post-autogamous cells will undergo the first cellular division and produce sexual progeny that will resume vegetative growth. The parental MAC is degraded into fragments that will eventually be diluted through cell divisions. **b** H3K27me3 and H3K9me3 immunostaining of non-injected cells (NI), and of cells transformed with *GFP-EZL1*[wt] or *GFP-EZL1*[H526A] RNAi-resistant transgenes upon control or *EZL1* RNAi during MAC development, at ~25 h after the onset of sexual events. The non-essential ND7 gene was used as RNAi control. Representative images (percentage in cell population is indicated, $n \geq 100$) are displayed. Overlay of Z-projections of magnified views of H3K27me3- or H3K9me3-specific antibodies (in cyan), of GFP (in green) and Hoechst (in red) are presented. Dashed white circles indicate the new developing MACs. The other Hoechst-stained nuclei are fragments from the old vegetative MAC. In control cells, H3K27me3 is detected in the fragments from the old vegetative MAC, while both H3K9me3 and H3K27me3 are detected in the new developing MACs. The scale bar is 10 μm. **c** Production of sexual progeny following control (CTL) or *EZL1* gene silencing in non-injected (NI) cells, *GFP-EZL1*[wt] and *GFP-EZL1*[H526A] transformed cells. Cells were starved in each medium to induce autogamy and, following 3–4 days of starvation, post-autogamous cells were transferred individually to standard growth medium to assess the ability of sexual progeny to resume vegetative growth. The total number of cells analyzed for each RNAi and the number of independent experiments (in parenthesis) are indicated. Transgene copy number per haploid genome (see Methods): 6.9 ($n = 30$) and 76.4 ($n = 58$) for *GFP-EZL1*[wt], 3.4 ($n = 30$), and 5.8 ($n = 60$) for *GFP-EZL1*[H526A]. **d** Quantification of the relative copy number of individual TEs (Anchois; RT12275; RT25765; RT31010; RT42890) as measured by qPCR from total genomic DNA extracted at the end of MAC development from non-injected (NI), *GFP-EZL1*[wt] and *GFP-EZL1*[H526A] transformed cells after control (CTL) or *EZL1* RNAi. Values were normalized to a MIC-specific, single copy genomic locus. The horizontal bars represent mean of biological replicates. Circles indicate the individual data points. Source data are provided as a Source Data file

end, we performed a time-course quantitative reverse transcription PCR (RT-qPCR) analysis of steady-state RNA levels upon knockdown of the Pgm endonuclease, which, similarly to *EZL1* knockdown, leads to complete retention of TE copies in the zygotic MAC genome[24,25,35]. In contrast to *EZL1* RNAi, we did not detect transcriptional upregulation of transposable elements upon *PGM* RNAi, as shown by RT-qPCR for three individual TEs (Fig. 4f and Supplementary Table 4). These results strongly suggest that retention of transposable elements in the MAC genome is not sufficient to trigger their transcriptional upregulation. Instead, upregulation of transposable element transcription can be considered a specific consequence of Ezl1 loss. Altogether, our analysis reveals that transposable elements undergo a global release from silencing upon *EZL1* knockdown.

**H3K9me3 and H3K27me3 co-occur on transposable elements**. To determine whether the TEs that are transcriptionally upregulated are targets of the Ezl1 methyltransferase, we devised a chromatin immunoprecipitation (ChIP) procedure optimized for developing *Paramecium* cells and measured local enrichment for H3K9me3 and H3K27me3 in wild-type cells at an early stage of development, followed by qPCR (ChIP-qPCR). ChIP-qPCR with H3K4me3 antibody was used as a control. Twelve individual TEs showed enriched occupancy of H3K27me3 and H3K9me3 compared to genes, regardless of whether these copies were transcriptionally upregulated upon *EZL1* RNAi (Fig. 5a). As expected, no enriched occupancy of H3K4me3 was detected for the TEs compared to control reference genes (Fig. 5a). Similar experiments performed in *PGM* knockdown cells showed the same enrichment profiles as those observed in wild-type cells, demonstrating that the Pgm endonuclease acts downstream of H3K27me3 and H3K9me3 deposition (Fig. 5b top panel). This result is in agreement with previous work showing that accumulation of H3K27me3 and H3K9me3 in the developing zygotic MAC is not altered upon *PGM* RNAi[25]. In contrast, a strong reduction in H3K9me3 and H3K27me3 enrichment could be observed on TEs upon *EZL1* knockdown (Fig. 5b bottom panel). Consistent with immunostaining experiments (Fig. 3 and ref. [25]), these data reveal a net loss of H3K9me3 and H3K27me3 on TEs upon *EZL1* RNAi.

The ChIP-qPCR analysis of individual loci indicates Ezl1-dependent co-occurrence of H3K27me3 and H3K9me3 on the same TE copies. To examine the chromatin states of transposable elements more globally, we focused our analysis on the distribution of H3K9me3, H3K27me3, and H3K4me3 marks by chromatin immunoprecipitation followed by deep-sequencing (ChIP-seq) in biological replicates upon *PGM* knockdown (Supplementary Table 5). The use of *PGM* knockdown cells allowed us to overcome the absence of synchrony in *Paramecium* cell cultures and the loss of histone marks as development and DNA elimination proceed. Indeed, *PGM* knockdown cells are blocked at a late developmental stage, after H3K9me3 and H3K27me3 are normally deposited and accumulated. ChIP-seq reads were mapped (see Methods) on the 61 TE consensus sequences[24]. Consistent with qPCR data, enrichment of H3K27me3 and H3K9me3 marks with respect to the input was observed on the TE consensus sequences, while there was no enrichment of H3K4me3 (Fig. 5c). Furthermore, there is a statistically significant correlation between the enrichment of H3K27me3 and the enrichment of H3K9me3 (Supplementary Fig. 5; Pearson correlation coefficient 0.923 and $R^2$ linear regression coefficient 0.853). Interestingly, all TE consensus sequences showed enriched occupancy of H3K27me3 and H3K9me3, regardless of whether the elements were transcriptionally upregulated upon *EZL1* knockdown or not (Fig. 5c and Supplementary Fig. 6). Our data support the conclusion that loss of H3K27me3 and H3K9me3 upon *EZL1* RNAi coincides with TE transcriptional derepression. However, releasing transcriptional repression does not necessary result in transcription activation. Altogether, our global analysis confirms and extends our loci-oriented qPCR results and reveals that H3K9me3 and H3K27me3 jointly decorate transposable elements.

## Discussion
Our findings show that the *Paramecium* Enhancer-of-zeste Ezl1 protein, although similar to human EZH2 SET domain in sequence and structure (Fig. 1), catalyzes methylation of both lysine 9 and lysine 27 on histone H3 through its SET domain, in vitro (Fig. 2) and in vivo (Fig. 3). *EZL1* knockdown results in simultaneous loss of H3K9me3 and H3K27me3 and massive transcriptional upregulation of transposable elements, with only a minor effect on protein-coding gene expression (Fig. 4 and Supplementary Fig. 4). Consistently, we also demonstrate that H3K9me3 and H3K27me3 co-occur on transposable elements in an Ezl1-dependent manner (Fig. 5). These observations reveal

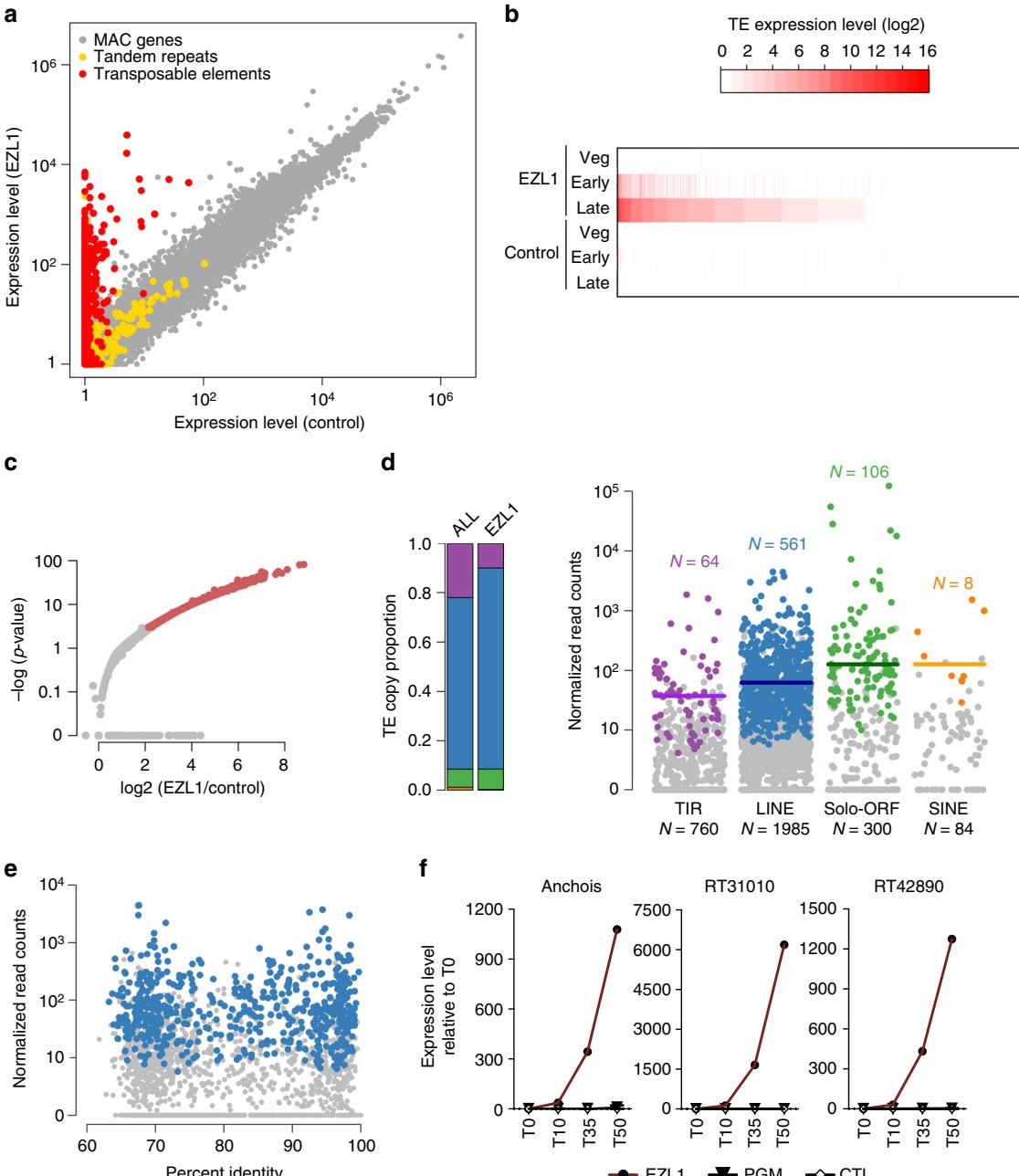

**Fig. 4** Loss of H3K27me3 and H3K9me3 leads to transcriptional derepression of transposable elements. **a** Scatter plot of the mean normalized expression level as measured by RNA-seq for *EZL1* and control samples of all elements ($n = 45,682$; 41,533 genes in gray, 3129 TE copies in red, 1020 tandem repeats in yellow). The non-essential *ICL7* gene was used as a RNAi control, because its silencing has no effect on sexual processes. **b** Heatmap of expression changes for TE copies at the three stages of the autogamy time-course (Veg, vegetative; Early; Late) in *EZL1* RNAi and *ICL7* (control) RNAi, ordered by *EZL1* Late expression level. **c** Volcano plot of TE copy expression ($n = 3129$) between *EZL1* and *ICL7* RNAi for the late stage. Red dots indicate significantly misregulation of 739 (23%) TE copies between *EZL1* and *ICL7* (control) RNAi (fold-change > 2 and *p*-value < 0.05). **d** Left, barplot showing the proportion of each TE family (TIR, purple; LINE, blue; solo-ORF, green; SINE, yellow) in all annotated TEs (ALL) and in the set of TEs whose expression is de-repressed upon *EZL1* RNAi (EZL1). Right, expression of individual elements from four different TE families for the late developmental stage. Each dot represents a single element. The differentially expressed copies are shown in the same colors used for the barplot. The horizontal bar represents the median expression level of the differentially expressed copies. The total number of copies in each family is indicated below and the number of differentially expressed copies for each family above the jitter plots. **e** Scatter plot for the 1985 LINE copies, showing the relation between identity with the TE consensus and *EZL1* expression level. Blue dots indicate the 561 differentially expressed LINE copies. **f** Expression of Anchois (TIR); RT31010 (LINE) and RT42890 (LINE) transposable elements at different time-points during development (T0; T10; T35; T50) upon *PGM*, *EZL1*, or control (*ICL7*) RNAi as measured by RT-qPCR. Values were normalized to *GAPDH* and are expressed as the fold-change to T0 (corresponds to the onset of sexual events) for each autogamy time-course experiment. Data represent the mean expression from two biological replicates for *PGM* RNAi. Data source are provided as a Source Data file

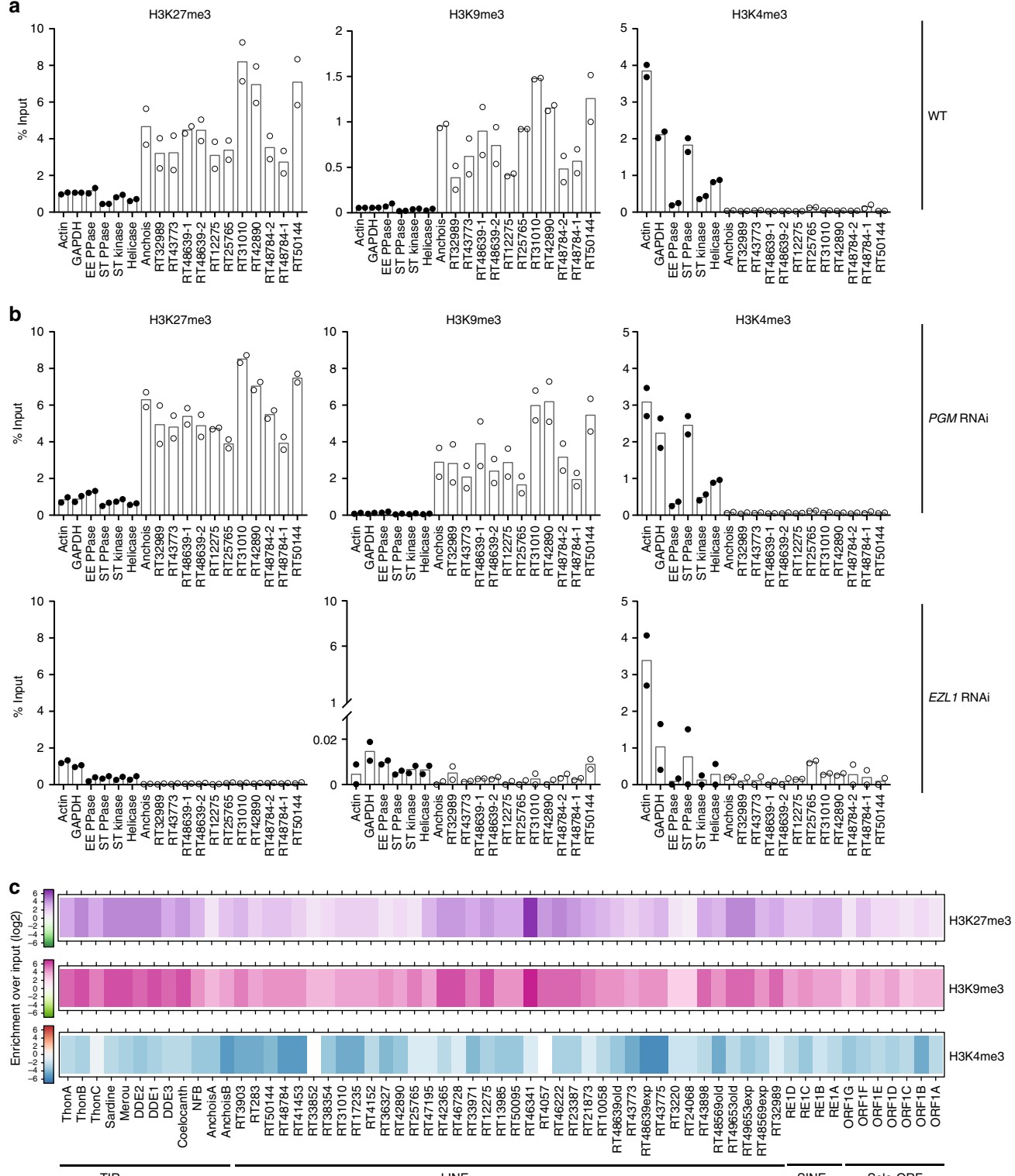

**Fig. 5** H3K27me3 and H3K9me3 mark the same transposable element copies in an Ezl1-dependent manner. **a** H3K9me3, H3K27me3, and H3K4me3 enrichment at individual copies from different TE families at an early stage of development (at 10 h after the onset of sexual events) in wild-type cells as measured by ChIP-qPCR. Quantitative data are expressed as the percentage of ChIP over Input. *ACTIN*, *GAPDH*, *EE PPASE*, *ST PPASE*, *ST KINASE*, and *HELICASE* are control genes (Supplementary Table 4 for accession numbers and primers). According to the RNA-seq analysis, Anchois (TIR), RT31010; RT42890; RT50144; RT48784–1; RT48639–1; RT48639–2 (LINEs) are significantly upregulated upon *EZL1* RNAi, whereas RT25765; RT12275; RT32989; RT43773; RT48784–1 (LINEs) are not. Circles (black for genes; white for TEs) indicate the individual data points. Horizontal bars represent the mean of biological replicates. **b** H3K9me3, H3K27me3, and H3K4me3 enrichment at the same individual TE copies at a late stage of development (50 h after the onset of sexual events) upon *PGM* (top panel) or *EZL1* (bottom panel) RNAi as measured by ChIP-qPCR. Data are represented as in **a**. **c** Heatmap of H3K27me3, H3K9me3, and H3K4me3 enrichment for 61 TE families determined by ChIP-seq of two biological replicates. Colors represent the fold-change (log2) in a given TE family for each histone mark relative to input. Source data are provided as a Source Data file

that an Enhancer-of-zeste protein mediates transcriptional silencing of transposable elements.

The SET domains of *Paramecium* Ezl1 and human EZH2 exhibit significant structural similarities (Fig. 1), yet Ezl1 displays distinct enzymatic properties. Although we cannot formally exclude the possibility that the effect of in vivo Ezl1 depletion on the methylation of a specific lysine is indirect, we believe that the most parsimonious explanation, given the body of experimental evidence obtained in this study, is that H3K9 and H3K27 are both recognized as substrates by the *Paramecium* Ezl1 protein. Indeed, we show that: (i) Ezl1 purified from *Paramecium* nuclear extracts is able to catalyze in vitro methylation of *Xenopus* and *Paramecium* histone H3 (Fig. 2 and Supplementary Fig. 2); (ii) this activity is severely compromised in the Ezl1$^{H526A}$ mutant, which carries a substitution in the highly conserved RXXNH cofactor-binding motif within the catalytic SET domain (Fig. 2 and Supplementary Fig. 2); (iii) K9 and K27 on *Paramecium* H3 are methylated in Ezl1-dependent manner (Fig. 2; Table 1; Supplementary Fig. 2; and Supplementary Data 1); (iv) Ezl1 in vitro methyltransferase activity is reduced using K9/27 double-mutant H3 compared to wild-type H3 or single mutants (Fig. 2 and Supplementary Fig. 2). (v) Ezl1 catalytic activity is necessary in vivo for the deposition of H3K9me3 and H3K27me3 (Fig. 3).

We noted that K23 is methylated in vitro in an Ezl1-dependent manner when lysines 9 and 27 are mutated and no longer substrates for methylation by the enzyme (Fig. 2 and Supplementary Data 1). Future work will be needed to determine whether this or additional methylations occur in an Ezl1-dependent manner in vivo and whether they are biologically significant.

In vitro methyltransferase assays performed with purified PRC2 have shown activity on both K9 and K27 of the histone H3 tail, albeit with an overwhelming preference for K27 (Fig. 2c and refs. [7–9]). In vivo, however, H3K9 methylation levels are not affected in the absence of PRC2[6,36,37]. The amino acid sequence of the H3 tail is highly conserved and the H3K9 and H3K27 lysines are typically embedded within a similar ARKS peptide sequence, making plausible the existence of common modifying enzymes. Interestingly, KDM7 is a dual demethylase with the ability to mediate the direct removal of methyl groups from both H3K9 and H3K27[38].

Altogether, our results show that Enhancer-of-zeste protein Ezl1 displays a broader specificity than EZH2, with both K9 and K27 on histone H3 as its main targets. In the future, it will be interesting to determine what properties, of Ezl1 itself or its associated complex subunits, confer broader specificity to this Enhancer-of-zeste homolog.

Our ChIP experiments show co-occurrence of H3K9me3 and H3K27me3 on the same TE copies. Loss of these marks upon *EZL1* RNAi coincides with TE transcriptional derepression. Altogether, our data demonstrate the function of an Enhancer-of-zeste protein in silencing genomic parasites such as transposable elements. This Ezl1-dependent silencing mechanism is broad and does not discriminate among the elements that it silences. Indeed, the genome-wide relaxation observed upon Ezl1 depletion involved all TE families, DNA transposons as well as non-LTR retrotransposons, but not all copies within a given family (Fig. 4). No correlation could be observed between the age of the copy and its expression level upon *EZL1* RNAi, indicating that other, unidentified factors may explain this difference. In contrast to our findings, previous work has uncovered only isolated examples of Enhancer-of-zeste involvement in controlling transposable and repetitive elements. The PRC2 pathway has been reported to repress the expression of two types of endogenous retroviruses in mouse embryonic cells[39]. A different mechanism, in which the retinoblastoma (Rb) protein utilizes an interaction with E2F1 to recruit

Enhancer-of-zeste-homolog 2 (EZH2) and establish H3K27me3, silences expression of genomic repeats, such as tandem repeats and retrotransposons, in mouse embryonic fibroblasts. However, the upregulation of repeat sequences in Rb mutants is rather weak and appears quite variable[40]. In the unicellular green alga *Chlamydomonas reinhardtii*, inactivation of an Enhancer-of-zeste homolog was shown to derepress the expression of one retrotransposon, but the effect was not analyzed genome-wide[41]. The scale of TE derepression we observe in *Paramecium* is remarkable: ~24% of the known TE copies show a 2 to > 500-fold-change in expression compared to control conditions. Such a massive level of derepression can be explained, at least in part, by the fact that germline TE copies are retained in the zygotic MAC genome upon H3K9me3 and H3K27me3 loss and are highly endoreplicated during development. This peculiarity of the biological system may thus allow detection of TE relaxation at the whole-genome scale, and offers an ideal model organism to further elucidate the underlying mechanisms.

How the Ezl1 protein is recruited to its target sites remains to be discovered. Previous work has shown that developmental-stage-specific small RNAs (sRNAs) produced by the Dicer-like proteins Dcl2 and Dcl3 are required for the deposition of H3K9me3 and H3K27me3 in the developing zygotic MAC[25]. We therefore propose that Dcl2/Dcl3-dependent sRNAs guide the deposition of H3K9me3 and H3K27me3 to homologous loci, which include TEs, in the developing zygotic MAC. The H3K27me3 and H3K9me3 marks would then promote the recruitment (or activation) of the Pgm endonuclease, leading to the elimination of the marked chromatin[23,42].

Heterochromatin formation via sRNA-dependent mechanisms has been observed in a variety of organisms[43]. For instance, the RNAi machinery directs silencing at repetitive centromeric regions in *S. pombe* in a process that involves H3K9 methylation. Nuclear pathways involving sRNAs and H3K9 methylation also repress repetitive elements and genes in different eukaryotes, including *Arabidopsis*, *Drosophila*, mice, worms, and humans[43]. Transcriptional silencing of repetitive genomic elements is thus typically ensured by H3K9me3 and is associated with constitutive heterochromatin. There are striking similarities between these sRNA-dependent heterochromatin formation mechanisms and sRNA-directed DNA elimination, except that, in *Paramecium*, it goes a step further with the physical elimination of sequences that are associated with H3K9me3 and H3K27me3.

From an evolutionary point of view, one important implication of our work is that the primary function of the ancient family of Enhancer-of-zeste proteins may originally have been to silence transposable elements. We propose an evolutionary scenario whereby ancestral Enhancer-of-zeste proteins, like the *Paramecium* Ezl1, catalyzed both H3K9me3 and H3K27me3, and played a role in the silencing of intragenomic parasites, such as transposable elements, through the concerted action of both of these histone marks. Enzymes dedicated to the catalysis of H3K9me3 may later have been co-opted to silence genomic repeats and Enhancer-of-zeste enzymes may then have evolved for lineage-specific roles, like the regulation of genes involved in development.

In favor of this hypothesis, PRC2 components are widespread in the major supergroups of eukaryotes (Opisthokonta (animals and fungi), Archaeplastida (red algae, green algae and plants), and Chromalveolata (ciliates such as *Tetrahymena* and *Paramecium*)), which strongly supports the idea that PRC2 subunits emerged in an ancestral unicellular eukaryote[41]. The phylogenetic distribution of H3K9me3-specific enzymes, the Su(var)3–9 homologs, seems instead to be restricted to unikonts (animals,

fungi, and Amoebozoa) and plants, according to ref. [44]. Yet the existence of H3K9me3 appears to predate the appearance of these K9-specific methyltransferases, based on observations made in *Chlamydomas*[41], diatoms[45], and ciliates[25,46,47]. In the diatom *P. tricornutum*, H3K27me3 was shown to occur on transposable elements together with H3K9me3[45], in agreement with the hypothesis of an evolutionarily ancient function for a dual H3K27/H3K9 methyltransferase in transcriptional repression of TEs.

Even though eukaryotic genomes are often described as containing two types of heterochromatin that serve largely distinct functions, there are examples in animals and fungi of an interplay between H3K27me3 and H3K9me3. For instance, H3K27me3 and H3K9me3 overlap at a subset of developmental genes in some cell types[15–17], suggesting that H3K27me3 and H3K9me3 cooperate to maintain gene silencing[48]. Similarly, domains where H3K9me3 and H3K27me3 co-occur have also been reported in the nematode *Caenorhabditis elegans*[11], and both marks are required for silencing of heterochromatic transgenes[49]. There are also some hints that Polycomb systems may interact with constitutive heterochromatin associated with H3K9me3, both in normal and mutant contexts. In the early mouse embryo, H3K27me2/3 accumulate at paternal pericentromeric heterochromatin after fertilization[50,51]. Perturbation of constitutive heterochromatin, through loss of H3K9me3 in pericentric heterochromatin in SUV39H knock-out mouse embryonic stem cells, is associated with new domains of H3K27me3 in patterns resembling those typical of H3K9me3[13]. In the filamentous fungus *N. crassa*, disruption of H3K9me3 causes redistribution of H3K27me2/3 to regions of constitutive heterochromatin, including centromeres[52]. Meanwhile, in the human fungal pathogen *C. neoformans*, H3K27me3 is lost from the telomeric regions it normally occupies and becomes redistributed to centromeric regions when a chromodomain protein that is part of the PRC2 complex in this organism is mutated[53]. Collectively, these examples reveal that H3K27me3 can relocate under some circumstances, resulting in cohabitation of both H3K9me3 and H3K27me3 histone marks in the same genomic regions. This suggests a widespread functional redundancy between H3K27me3 and H3K9me3 pathways. Our current findings provide an ideal foundation for determining the extent of this redundancy and untangling its mechanisms.

## Methods

***Paramecium* strains, cultivation, and autogamy**. Unless otherwise stated, all experiments were carried out with the entirely homozygous strain 51 of *P. tetraurelia* (WGP). Cells were grown in wheat grass powder (WGP) (Pines International) infusion medium bacterized the day before use with *Klebsiella pneumoniae*, unless otherwise stated, and supplemented with 0.8 mg/mL β-sitosterol (Merck). Cultivation and autogamy were carried out at 27 °C as described[54,55].

**Injection of tagged *EZL1* transgenes**. The previously published *GFP-EZL1* transgene[25] was used to generate RNAi-resistant derivatives. The original *Pst*I-*Hpa*I restriction fragment of the *EZL1* coding sequence that comprises the RNAi target region (*EZL1-1* in ref. [25]) was replaced by a synthetic DNA sequence (Eurofins Genomics) designed to maximize nucleotide sequence divergence with the endogenous genomic locus without modifying the amino acid sequence of the encoded Ezl1 protein. Within this synthetic DNA fragment, *Spe*I and *Blp*I restriction sites were introduced at positions 1393 and 1685 of the *EZL1* coding sequence (GSPATG00037872001/PTET.51.1.G1740049) on either side of the highly conserved RXXNH motif within the SET domain (position 1636 to 1650). To generate the *GFP-EZL1^H526A* mutant transgene, the *Spe*I-*Blp*I fragment was replaced by a synthetic DNA sequence (Eurofins Genomics), in which the histidine-coding triplet AGG from nucleotide 1648 to 1650 was replaced by GCT. The resulting missense mutation results in a histidine to alanine substitution in position 526 of the Ezl1 protein (H526) within the RXXNH motif[3,31,32]. Plasmids expressing *FLAG-HA-EZL1* and *FLAG-HA-EZL1^H526A* are the same as the *GFP-EZL1* plasmids, except that a 3xFLAG (DYKDDDDK-DYKDDDDK-DYKDDDDK)—HA (YPYDVPDYA) tag, that was codon-optimized for the *P. tetraurelia* genetic code, is fused to *EZL1* in place of GFP. Plasmids were linearized with *Xmn*I and microinjected into the MAC of vegetative 51 cells.

For genetic complementation experiments, two transformed clones for each RNAi-resistant transgene (*GFP-EZL1* or *GFP-EZL1^H526A*) were chosen. The relative copy number of the transgenes was measured by qPCR from total genomic DNA extracted from vegetative cells and values were normalized to an endogenous gene (PTET.51.1.G1740048) (transgene copy number per haploid genome). No lethality was observed in the post-autogamous progeny of injected cells, indicating that the GFP-Ezl1 and GFP-Ezl1^H526A fusion proteins did not interfere with normal progression of autogamy in normal conditions (Fig. 3).

**Gene silencing experiments**. Plasmids used for T7Pol-driven dsRNA production in silencing experiments were those previously described[25]: *ND7*, *ICL7a*, *EZL1-1*, and *PGM-1*. Preparation of silencing medium and RNAi during autogamy were performed as described in ref. [25]. Lethality of post-autogamous cells after RNAi was assessed by transferring 30–60 individual post-autogamous cells to standard growth medium. Cells with a functional new MAC were identified as normally growing survivors unable to undergo a novel round of autogamy if starved after ~8 divisions.

**Antibodies, dot-blot, and competition assays**. Polyclonal rabbit anti-GFP (1:5000, Abcam, ab290) and monoclonal mouse anti-FLAG (1:1000, ThermoFisher Scientific, MA1–91878) antibodies were used for western blot analysis. Rabbit monoclonal anti-H3K4me3 antibodies were purchased from Merck Millipore (Billerica, MA, USA; catalog number #04–745, batch #2739249). Considering the fact that *Paramecium* histone H3 displays some amino acid differences with respect to mammalian histone H3 (Supplementary Fig. 2), polyclonal rabbit antibodies were raised to peptide sequences of the *Paramecium* H3 protein with Eurogentec (Seraing, Belgium): MSRDIQLARRIRGERF for anti-H3, KQTARK(me3)STAGNK for anti-H3K9me3 and TKAARK(me3)TAP for anti-H3K27me3 antibodies. Specificity of the custom made anti-modification antibodies was validated by dot-blot and competition assays (Supplementary Fig. 7).

**Indirect immunofluorescence**. Cells were fixed as previously described[25] with some modifications. Briefly, cells were collected in centrifuge tubes and fixed for 30 min in Solution I (10 mM EGTA, 25 mM HEPES, 2 mM MgCl₂, 60 mM PIPES pH 6.9 (PHEM 1 ×); 1% formaldehyde (Sigma-Aldrich), 2.5% Triton X-100, 4% Sucrose), and for 10 min in solution II (PHEM 1 ×, 4 % formaldehyde, 1.2% Triton X-100, 4% Sucrose). Following blocking in 3% bovine serum albumin-supplemented Tris buffered saline-Tween 20 0.1 % (TBST) for 10 min, fixed cells were incubated overnight at room temperature with primary antibodies as follows: rabbit anti-H3K9me3 (1:200), rabbit anti-H3K27me3 (1:1000). After two washes in TBST, cells were labeled with Alexa Fluor 568-conjugated goat anti-rabbit IgG (Invitrogen, catalog number #A-11036) at 1:500 for 1 h, stained with 1 μg/mL Hoechst for 5–10 min, washed in TBST and finally mounted in Citifluor AF2 glycerol solution (Citifluor Ltd, London). Images were acquired using a Zeiss LSM 780 laser-scanning confocal microscope and a Plan-Apochromat 63 × /1.40 oil DIC M27 objective. Z-series were performed with Z-steps of 0.35 μm.

**DNA extraction and real-time quantitative PCR**. DNA samples were typically extracted from 200–400 mL cultures of exponentially growing cells at 1000 cells/mL or of autogamous cells at 2000–4000 cells/mL[24]. DNA concentration was quantified with the Qubit dsDNA BR Assay Kit (Thermo Fisher Scientific) and stored at 4 °C. Real-time quantitative PCR was performed using the ABsolute QPCR SYBR Green Capillary Mix (Thermo Fisher Scientific) on the LightCycler 1.5 thermal cycling system (Roche). DNA amount of individual TEs was normalized to a MIC-specific, single copy genomic locus (IES 51A1835). Primers are listed in Supplementary Table 4.

**RNA extraction, RT-qPCR, and RNA sequencing**. For total RNA extraction, 200–400 mL cultures of exponentially growing cells at 1000 cells/mL or of autogamous cells at 2000–4000 cells/mL were centrifuged and flash-frozen in liquid nitrogen prior to TRIzol (Invitrogen) treatment, modified by the addition of glass beads for the initial lysis step. RNA concentration was quantified with the Qubit RNA HS Assay Kit (Thermo Fisher Scientific).

RNA samples were reverse-transcribed with RevertAid H Minus Reverse Transcriptase (Thermo Fisher Scientific) using random primers (Thermo Fisher Scientific) according to the manufacturer's instructions. Real-time quantitative PCR was performed using the ABsolute QPCR SYBR Green Capillary Mix (Thermo Fisher Scientific) on the LightCycler 1.5 thermal cycling system (Roche). Expression levels were normalized to the *GAPDH* gene and to the T0 developmental time point within each condition with the ΔΔCt method. Primers are listed in Supplementary Table 4.

For RNA-seq libraries, poly-A + RNA was extracted at different time-points during autogamy (Veg; T0; T5; T10; T20; T35; T50) from *ICL7* and *EZL1* RNAi samples described in ref. [25] using the FastTrack MAG mRNA isolation kit (Thermo Fisher Scientific) following the manufacturer's instructions. Strand-oriented Illumina libraries were made as described in ref. [34]. HiSeq paired-end sequencing

was performed on the samples, yielding at least $2 \times 30$ million reads, 100 nt in length, for each of the samples (Supplementary Table 1).

**Chromatin immunoprecipitation.** In all, $5–8 \times 10^6$ autogamous cells were collected by centrifugation and cross-linked with 1% methanol-free formaldehyde (Thermo Fisher Scientific) PHEM $1 \times$ (10 mM EGTA, 25 mM HEPES, 2 mM MgCl$_2$, 60 mM PIPES pH 6.9) for 10 min at room temperature. After quenching with 0.125 M glycine, cells were washed and pelleted. Cells were then lysed with a Potter-Elvehjem homogenizer in lysis buffer (0.25 M sucrose, 10 mM Tris pH 6.8, 10 mM MgCl$_2$, 0.2% Nonidet P-40, 1 mM PMSF (Sigma-Aldrich), 4 mM benzamidine (Sigma-Aldrich); 1x Complete EDTA-free Protease Inhibitor Cocktail tablets (Roche)). Following the addition of 2.5 volumes of washing solution (0.25 M sucrose, 10 mM MgCl$_2$, 10 mM Tris pH 7.4, 1 mM PMSF (Sigma-Aldrich), 4 mM benzamidine (Sigma-Aldrich), 1x Complete EDTA-free Protease Inhibitor Cocktail tablets (Roche)), the nuclei-containing pellet was collected by centrifugation and resuspended at $2 \times 10^6$ cells/mL in RIPA buffer without sodium dodecyl sulfate (SDS) (50 mM Tris pH 8, 150 mM NaCl, 10 mM EDTA, 1% Nonidet P-40, 0.5%, sodium deoxycholate, 1 mM PMSF (Sigma-Aldrich), 4 mM benzamidine (Sigma-Aldrich), 1x Complete EDTA-free Protease Inhibitor Cocktail tablets (Roche)), flash-frozen in liquid nitrogen as 200 μL aliquots and stored at −80 °C. After addition of SDS to a final concentration of 1%, chromatin was sonicated with Bioruptor (Diagenode) to reach a fragment size of 200–500 bp. DNA concentration was quantified with the Qubit dsDNA BR Assay Kit (Thermo Fisher Scientific). Chromatin corresponding to 13 μg of DNA was diluted 10-fold in RIPA buffer without SDS complemented with protease inhibitors and incubated overnight at 4 °C with 4 μg of H3K9me3 or H3K27me3 antibodies or with 4 μL of the H3K4me3 monoclonal antibody. As input control, 10% of sheared chromatin was taken aside. Antibody-bound chromatin was recovered after incubation for ~7 h at 4 °C with Magna ChIP Protein A + G Magnetic Beads (16–663, Millipore). Beads were washed twice with Low salt buffer (0.1% SDS, 1% Triton, 2 mM EDTA, 20 mM Tris pH 8, 150 mM NaCl), twice with High salt buffer (0.1% SDS, 1% Triton, 2 mM EDTA, 20 mM Tris pH 8, 500 mM NaCl), once with LiCl wash buffer (10 mM Tris pH 8.0, 1% Na-deoxycholate, 1% Nonidet P- 40, 250 mM LiCl, 1 mM EDTA) and once with TE 50 mM NaCl (50 mM Tris pH 8.0, 10 mM EDTA, 50 mM NaCl). Chromatin was eluted with TE 1% SDS (50 mM Tris pH 8.0, 10 mM EDTA, 1% SDS) at 65 °C for 45 min. ChIP-enriched samples and inputs were then reverse-cross-linked at 65 °C overnight and treated with RNase A and proteinase K. DNA was extracted with phenol, precipitated with glycogen in sodium acetate and ethanol and resuspended in deionized distilled water. Enrichment compared to input was analyzed by qPCR. qPCR was performed using the ABsolute QPCR SYBR Green Capillary Mix (Thermo Fisher Scientific) on the LightCycler 1.5 thermal cycling system (Roche). qPCR primers are listed in Supplementary Table 4. For RT48784-1 and RT48784-2, sequencing of PCR products confirmed that each set of primers specifically amplified a single TE copy. Approximately 5 ng of DNA were used for ChIP-seq library preparation. Truseq Illumina adapters were ligated and the libraries have been sized to 200–500 pb using a Beckman SPRI-TE automate and a spriworks fragment library kit (Beckman coulter). Ligation were amplified (15 cycles) with Kapa Hifi DNA polymerase (Kapa Biosystem). Libraries were pooled and sequenced on an Illumina NextSeq500 instrument, according to the manufacturer recommendations, in a Paired-End $2 \times 80$ pb run (NextSeq500/ 550 Mid Output Kit v2 (150 cycles)).

**Nuclear extraction and immunoprecipitation.** To prepare nuclear extracts, autogamous cells were lysed with a Potter-Elvehjem homogenizer in lysis buffer (10 mM Tris pH 6.8, 10 mM MgCl$_2$, 0.2% Nonidet P-40, 1 mM PMSF (Sigma-Aldrich), 4 mM benzamidine (Sigma-Aldrich), 1x Complete EDTA-free Protease Inhibitor Cocktail tablets (Roche)). Following the addition of 2.5 volumes of washing solution (0.25 M sucrose, 10 mM MgCl$_2$, 10 mM Tris pH 7.4, 1 mM PMSF (Sigma-Aldrich), 4 mM benzamidine (Sigma-Aldrich), 1x Complete EDTA-free Protease Inhibitor Cocktail tablets (Roche)), the nuclei-containing pellet was collected by centrifugation and incubated in 1 volume of nuclear extraction buffer $2 \times$ (100 mM Hepes (pH 7.8), 100 mM KCl, 600 mM NaCl, 0.2 mM EDTA, 20% Glycerol, 2 mM DTT, 0.02% Nonidet P-40, 2 mM PMSF, 2x Complete EDTA-free Protease Inhibitor Cocktail tablets (Roche)) at 4 °C overnight. The salt-extractable fraction was recovered following centrifugation at $10,000 \times g$ at 4 °C for 3 min. Protein concentration was quantified with the Qubit Protein Assay Kit (Thermo Fisher Scientific).

For immunoprecipitations of GFP fusion proteins, 400 μg of nuclear protein extracts at 12 h after the onset of sexual events obtained from *Paramecium* transformed cells, in which endogenous Ezl1 protein expression was specifically depleted through RNA interference, was used for GFP-Trap immunoprecipitation. Briefly, nuclear extracts were incubated for 1 h at 4 °C with GFP-Trap_M magnetic beads (Chromotek). Beads were washed three times in Wash Buffer (10 mM Tris/ Cl pH 7.5, 600 mM NaCl, 0.5 mM EDTA, 0.01% Nonidet P-40).

For tandem affinity immunoprecipitation of 3xFLAG-HA tagged proteins, 1 mg of nuclear extracts was incubated overnight at 4 °C with anti-FLAG M2 magnetic beads (M8823, Sigma). Beads were washed five times with TEGN buffer (20 mM Tris pH 8, 0.1 mM EDTA, 10% Glycerol, 300 mM NaCl, 0.01% NP40). After elution with the FLAG peptide (F4799, Sigma), the eluate was further affinity-purified on anti-HA antibody-conjugated agarose (A2095, Sigma) overnight at 4 °C and eluted with the HA peptide (I2149, Sigma) overnight at 4 °C.

**Histone methyltransferase assays.** GFP-Trap IP beads were resuspended in 50–100 μL of HMT buffer $1 \times$ (50 mM Tris-HCl pH 8.5, 2.5 mM MgCl$_2$) and aliquots of 1 to 5 μL were used. For histone methyltransferase assays with tandem affinity purified FLAG-HA IP, 2 μL of the final eluate was used. Beads (or eluates) were incubated with either recombinant *Xenopus* histone octamers (rOct, 1 μg), reconstituted oligonucleosomes (2 μg), prepared as previously described[56], or recombinant *Paramecium* histone H3/*Xenopus* histone H4 tetramers (1 μg) in the presence of 4 mM DTT and 2 μM $^3$H-labeled S-adenosylmethionine, in a volume of 25 μL per reaction, for 30 min at 30 °C. *Paramecium* histone H3 (H3P1[57]), encoded by a synthetic gene (Eurofins Genomics) adapted to the universal genetic code, was expressed and purified from *E. coli* then used to assemble histone tetramers by salt dialysis as described[56]. Reactions were stopped by addition of sodium dodecyl sulfate polyacrylamide gel electrophoresis (SDS-PAGE) sample buffer and analyzed by SDS-PAGE, transfer to polyvinylidene difluoride membranes, Coomassie staining, and autoradiography. Affinity-purified, reconstituted mammalian PRC2 core complex (EED, EZH2, SUZ12 and RbAp48) was prepared as previously described[58] and used as a positive control.

**Quantitative label-free mass spectrometry.** Full-length 3xFLAG-HA-tagged Ezl1$^{wt}$ and the catalytic mutant Ezl1$^{H526A}$ were immunoprecipitated using a FLAG-HA double-affinity purification from nuclear extracts prepared 12 h after the onset of sexual events in transformed *Paramecium* cells, depleted for the endogenous Ezl1 protein in the case of FLAG-HA-Ezl1$^{H526A}$. In vitro histone methyltransferase assays were performed with immunoprecipitated Ezl1$^{wt}$ and Ezl1$^{H526A}$ proteins from three biological replicates (independent nuclear extracts) and with recombinant histone tetramers (2 μg per reaction), assembled with wild-type *Paramecium* histone H3 (H3P1[57]) histones as substrates and 30 μM S-adenosylmethionine as methyl donor, in the presence of 4 mM DTT, in a volume of 25 μL per reaction, for 1 h at 30 °C. The same reactions with histone tetramers only were performed in triplicate as negative controls. The reaction products were separated on a 15% SDS-PAGE gel and stained with colloidal coomassie blue (LabSafe Gel Blue™, AGRO-BIO) without methanol and acetic acid.

The band corresponding to histone H3 was excised for each sample and washed, and proteins were reduced with 10 mM DTT prior to alkylation with 55 mM iodoacetamide. After washing and shrinking of the gel pieces with 100% acetonitrile, propionylation was performed. The shrink gel bands were chemically derivatized by treatment with propionic anhydride before and after trypsin digestion. Briefly, this reaction mixture was created using ¾ propionyl anhydride (Sigma-Aldrich) and ¼ isopropanol. Twenty microliters of propionylation reagent were added to each band adjusted with ammonium hydroxide to pH 8 and allowed to react at 51 °C for 20 min (two times) and reduced to dryness using a SpeedVac concentrator for removal of reaction remnants before trypsin digestion. Digestions were done overnight in 25 mM ammonium bicarbonate at 30 °C, by adding 10 μL endoproteinase (10 ng/μL) trypsin/LysC (Promega). Peptides were extracted by ultrasonic with extraction solution (60% acetonitrile/5% formic acid). The extraction was dried in a vacuum concentrator at room temperature. A second round of propionylation was performed to propionylate the newly created peptide N-termini. Sample were then loaded onto a homemade C18 StageTips. Peptides were eluted from beads by incubation with 40/60 MeCN/H$_2$O + 0.1% formic acid. The peptides were dried in a Speedvac and reconstituted in 10 μL 2/98 MeCN/ H$_2$O + 0.3% trifluoroacetic acid (TFA) prior to liquid chromatography–tandem mass spectrometry (LC-MS/MS) analysis.

Peptide samples (5 μL) were chromatographically separated using an RSLCnano system (Ultimate 3000, Thermo Scientific) coupled online to a Q Exactive HF-X with a Nanospray Flex ion source (Thermo Scientific). Peptides were first trapped on a C18 column (75-μm inner diameter × 2 cm; nanoViper Acclaim PepMap™ 100, Thermo Scientific), with buffer A at a flow rate of 2.5 μL/min over 4 min. Separation was then performed on a C18 column (75-μm inner diameter × 50 cm; nanoViper C18, 2 μm, 100 Å, Acclaim PepMap™ RSLC, Thermo Scientific) regulated to a temperature of 50 °C with a linear gradient of 2 to 35% buffer B (100% MeCN and 0.1 % formic acid) at a flow rate of 300 nL/min over 91 min. MS full scans were performed in the ultrahigh-field Orbitrap mass analyzer in ranges $m/z$ 375–1500 with a resolution of 120,000 at $m/z$ 200. The top 20 intense ions were subjected to Orbitrap for further fragmentation via high-energy collision dissociation (HCD) activation and a resolution of 15,000 with the intensity threshold kept at $1 \times 10^5$. We selected ions with charge state from $2 +$ to $6 +$ for screening. Normalized collision energy (NCE) was set at 27 and the dynamic exclusion of 20 s.

For identification, the data were searched against the UniProt *Paramecium tetraurelia* (downloaded on 10 September 2018 and containing 39,463 sequences) database using Mascot (version 2.5.1). Enzyme specificity was set to trypsin and a maximum of five-missed cleavage sites were allowed. Oxidized methionine, N-terminal acetylation and Propionylation, Methyl lysine, Dimethyl lysine, Trimethyl lysine, Propionyl lysine, Methyl-Propionyl lysine, and carbamidomethyl cysteine were set as variable modifications. Maximum allowed mass deviation was set to 10 ppm for monoisotopic precursor ions and 0.02 Da for MS/MS peaks. The resulting files were further processed using myProMS v3.6 (https://github.com/bioinfo-pf-curie/myproms)[59], where we fixed the false discovery rate for peptide identification to <5%. All peptide identifications were manually inspected to ensure the quality of the analysis.

To quantify the modified peptides, we extracted from the MS survey of nano-LC-MS/MS raw files the extracted ion chromatogram (XIC) signal of the well characterized tryptic peptide ions using Skyline (version 4.1) MacCoss Lab Software, Seattle, WA; https://skyline.ms/project/home/sofware/Skyline/begin.view). The peptide XIC areas were log2 transformed and the mean log2- area was normalized by the mean area of peptide A[Propionyl (N-term)]RTK[Propionyl (K)]QTAR using the R statistical computing environment version 3.1.0. On each peptide a linear model was used to estimate the mean and the 5% confidence interval of each condition. The associated ratio and $p$-value have been computed thanks to a two-sided $t$-test with the Welch approximation of the number of degrees of freedom.

**Protein structure modeling**. ClustalW[60] was used for sequence alignment of target human EZH2 (Q15910) sequence and template Ezl1 (PTET.51.1.G1740049) sequence. Modeller v 9.19[61] was used for comparative homology modeling of Ezl1 SET domain using the same criteria as in ref. [62]. The structure of EZH2 in interaction with the peptide substrate analog and the S-adenosylhomocysteine (SAH) cofactor (PDB ID: 5HYN) was used as a template. Pymol molecular graphics system (Version 1.7.2.1 Schrödinger, LLC) was used for comparative structural analysis between reference and modeled structures. Model quality was analyzed using PROCHECK[63].

**RNA-seq data analysis**. This study used the *Paramecium tetraurelia* strain 51 MAC (ptetraurelia_mac_51.fa) and MAC + IES (ptetraurelia_mac_51_with_ies.fa) reference assemblies, *P. tetraurelia* gene annotation version 2[34] and *P. tetraurelia* IES annotation[35]. TE copies larger than 100 nt and not overlapping genes were identified using the TEAnnot module of the REPET-2.5 package[64] on the *P. tetraurelia* MIC contigs[24], using a manually curated library of 61 TE consensus sequences[24]. TEannot provides the percent identity between TE copies and the consensus (used as a proxy for the age of the copy). Tandem repeats not overlapping genes or TE were identified on MIC contigs larger than 500 nt using Tandem Repeat Finder[24,65]. Developmental genes were identified by RNA-seq DE analysis of wild-type *P. tetraurelia* strain 51[34]. All reference genomes and fasta librairies are available from ParameciumDB (https://paramecium.i2bc.paris-saclay.fr/download/).

After removal of contaminants and rDNA by mapping the 100 nt paired-end reads with bwa (default parameters), the RNA-seq reads were mapped using TopHat (v. 2.0.12) on the MAC and the MAC + IES reference genomes, allowing two mismatches and requiring unique mapping in order to assure high specificity. We used htseq-count (v. 0.6.0) to count fragments on the genes (and on the IESs they contain). The reads that did not map on MAC or MAC + IES references were mapped using TopHat (−multihits = 1000 and allowing two mismatches) on the MIC contigs. By allowing multiple hits, we avoid underestimating the expression of repeated elements. Fragments mapped to TE copies and tandem repeats were counted with htseq-count.

For differential expression analysis, fragment counts were normalized using the 45,682 elements (genes, TE copies and repeats) and DESeq2 (v. 1.14−1) in the R statistical computing environment (R project v 3.3.2). PCA was carried out using the R package FactoMineR (v. 1.34). Differential expression of genes and TE copies was carried out with DESeq2, requiring a $p$-value < 0.05 and a fold-change > 2. Gene heatmaps were made with the heatmap.2 and hclust R methods.

**ChIP-seq data analysis**. ChIP-seq data were demultiplexed using CASAVA (v1.8.2) and bcl2fastq2 (v2.18.12). Illumina adapters were removed using cutadapt (1.12), keeping only reads with a minimal length of ten nucleotides. Paired-end 80 nt reads were mapped (bowtie2.3.3) on expected contaminants (mitochondrial genomes, ribosomal DNA, and bacterial genomes). PCR duplicates and low quality reads (sickle v1.200 -q 20) were removed from the unmapped reads. Supplementary Table 5 shows the mapping statistics on *Paramecium* genomes (MAC: ptetraurelia_mac_51.fa; MIC: ptetraurelia_mic2.fa) using bowtie2 v2.3.3 with mapping quality >20. The data were also mapped on the 61 *Paramecium* TE consensus sequences[24] (bowtie2.3.3 default parameters) without any mapping quality restriction. The coverage on each TE was calculated using bedtools (v2.17.0) then normalized by a sequencing scale factor method.

**Reporting summary**. Further information on research design is available in the Nature Research Reporting Summary linked to this article.

## Data availability

RNA-seq and ChIP-seq datasets generated for this study have been deposited at the ENA with Study Accession number PRJEB21555. Supplementary Tables 1 and 5 describe the samples, mapping statistics, and provide accession numbers for each sample. The mass spectrometry proteomics data have been deposited to the ProteomeXchange Consortium via the PRIDE[66] partner repository with the data set identifier PXD012170. All other relevant data supporting the key findings of this study are available within the article and its Supplementary Information files or from the corresponding author upon reasonable request. The source data underlying Figs. 2–5 and Supplementary Figs. 2 and 7 are provided as a Source Data file. A reporting summary for this article is available as a Supplementary Information file.

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

## Acknowledgements

We thank Slimane Ait-Si-Ali and Michel Werner for critical reading of the manuscript. Special thanks to Slimane Ait-Si-Ali for advice for tandem affinity purification experiments, to Maoussi Lhuillier-Akakpo for her help in setting up the ChIP protocol, and to members of the Duharcourt lab for stimulating discussions. This work was supported by the Agence Nationale de la Recherche (ANR-14-CE10–0005–04 'PIGGYPACK') to S.D. and O.A. and the salary of E.E. was provided by this grant. Work in the laboratory of S.D. is supported by intramural funding from the CNRS, the Fondation de la Recherche Médicale (Equipe FRM DEQ20160334868), LABEX "Who am I?" (ANR-11-LABX-0071_WHOAMI). A.F. was recipient of Ph.D fellowships from 'Ministère de l'Enseignement Supérieur et de la Recherche' and 'Fondation de la Recherche Médicale', and of a postdoc transition fellowship from LABEX "Who am I?" supported by the ANR-11-LABX-0071_WHOAMI and the ANR-11-IDEX-0005–02. C.M.P. is recipient of a Ph.D fellowship from 'Ministère de l'Enseignement Supérieur et de la Recherche". D.H. is supported by a postdoctoral fellowship from the Fondation pour la Recherche Medicale (SPF20150934266). Work in the laboratory of R.M. is supported by a 'Programme Labellise' grant from the Fondation ARC pour la Recherche sur le Cancer. This work was supported by "Région Ile-de-France" and "Fondation pour la Recherche Médicale" grants to D. L. We acknowledge the ImagoSeine facility, member of the France BioImaging infrastructure supported by the ANR-10-INSB-04. The sequencing benefited from the facilities and expertize of the high-throughput sequencing platform of I2BC.

## Author contributions

A.F., C.M.P. performed most experiments with help from A.H. O.A., E.E., L.S. performed annotation, RNA-seq, and ChIP-seq analyses. D.H., T.K., R.M. contributed to the in vitro methyltransferase assays. K.G. performed the protein structure modeling. B.L. carried out the mass spectrometry experimental work and D.L. supervised MS and data analysis. S.D. wrote the manuscript and supervised the project.

## Additional information

**Competing interests:** The authors declare no competing interests.

