## [Peer Review File · Nature Communications]

Reviewers' comments:

Reviewer #1 (Remarks to the Author):

In this study, the authors showed that the Paramecium Enhancer-of-zest protein Ezl1 has a methyltransferase activity for both histone H3 lysine 9 (K9) and lysine 27 (K27) in vitro and in vivo. They also showed that Ezl1 is necessary for silencing many transposons and both H3K9me3 and H3K27me3 are accumulated on at least some of the transposons in Ezl1 dependent manner. Finally, from these results they provided an interesting discussion about the ancestral role of H3K27 methylation. I believe this is a novel and an important study that is worth to be published in Nature Communications but I think the following points should be clarified.

In this study, the authors showed that the Paramecium Enhancer-of-zest protein Ezl1 has a methyltransferase activity for both histone H3 lysine 9 (K9) and lysine 27 (K27) in vitro and in vivo. They also showed that Ezl1 is necessary for silencing many transposons and both H3K9me3 and H3K27me3 are accumulated on at least some of the transposons in Ezl1 dependent manner. Finally, from these results they provided an interesting discussion about the ancestral role of H3K27 methylation. I believe this is a novel and an important study that is worth to be published in Nature Communications but I think the following points should be clarified.

Related to the experiments shown in Figure 1:

- 1) Is the sequence of histone H3 (especially its N-terminal tail) of Paramecium identical to that of Xenopus and human? If not, some discussion is necessary to justify that the assays using Paramecium Ezl1 with vertebrate histones/nucleosomes are valid.
- 2) The differences between lanes 7/8 and 9 of Figure 1C might not be convincingly large. Are these statistically significant?
- 3) Some activity was detected in the lane 9 of Figure 1C. This might suggest that Ezl1 has a methyltransferase activity for some residues other than K9 and K27 of H3. I believe some discussion for this point is necessary in the text.

Related to the experiments shown in Figure 2:

- 4) The maternal MAC contains H3K27me3 in the control cells and it is dependent on the catalytic activity of Ezl1. Curiously, H3K9me3 is not in the maternal MAC even in the control cells. These suggest that Ezl1 catalyses only H3K27me3 in the maternal MAC but both H3K9me3 and H3K27me3 in the new MAC. First, I believe this point should be discussed in the text. Second, I think it is important to purify EGFP-Ezl1 complex from cells before the new MAC development and perform a methyltransferase assay like in Figure 1 to ask if it shows preference for H3K27.

Related to the experiments shown in Figure 4:

- 5) Are the PCR primers for ChIP-qPCR designed to specifically amplify a particular copy of a transposon family or to amplify a whole transposon family? If the latter is the case, it would not be possible to conclude that H3K9me3 and H3K27me3 co-occur on a same copy of a transposon.

Related to the last section of Discussions:

- 6) Does Paramecium have homologs of PRC2 subunits beside Ezl1? If yes, it would be interesting to discuss those proteins as well.

Reviewer #2 (Remarks to the Author):

In this manuscript, the authors report an SET domain containing protein, Ezl1, from the unicellular eukaryote Paramecium tetraurelia, is a dual methyltransferase that catalyzes trimethylation of histone H3 on both K27 and K9 in vitro and in vivo. In developing macronuclei, K9 and K27 trimethylation co-occur on transposable elements (TE) and inactivation of Ezl1 abolishes both of the two modifications and results in global de-repression of these repetitive sequences. The selling point is that H3K9me3 and H3K27me3 have a common ancestor and function together at constitutive heterochromatin, the latter was only reported at the pericentric heterochromatin in mammalian cells when DNA methylation is absent.

Overall, this study is largely descriptive and not thorough enough to be a strong candidate for Nature Communication.

Major concerns:

1. It is not totally surprising that an enzyme can do both H3K9 and H3K27 methylation. As the authors cited, PRC2 is known to display minor activities at H3K9. Moreover, G9a and GLP, the two major H3K9 dimethylase in mammals are also known to display some activities at H3K27. Thus, the potential dual substrate-specificity of Ezl1 does not appear to be novel enough.
2. The quality of purified Ezl1 is not satisfying. As the authors acknowledge, they cannot rule out the contribution of contaminant proteins in their in vitro activity assays. I agree that this is a valid concern. First of all, there is not silver staining of the purified protein or protein complex to demonstrate the purity of their preparation. Secondly, their catalytic dead mutant is not dead (Fig. 1A) and their enzyme remain active at K9/K27 double mutant substrate (Fig. 1C). This certainly raise a possibility that some of the HMT activities observed in their in vitro assay come from contaminant proteins.
3. Enhancer of Zeste and its homologues proteins from mammals, fly, *C. elegans* and even thermophilic fungi exist in multi-subunit complex and do not exert HMT activity alone. Does Ezl1 from *Paramecium tetraurelia* also forms a complex with other proteins? This should be answered since the authors have already purified it from the cells.
4. Does *Paramecium tetraurelia* have other enzymes responsible for H3K9me3 and H3K27me3?
5. What is the extent of colocalization of H3K9me3 and H3K27me3 at the genome-wide level? ChIP-seq experiments will be helpful to address this and also to demonstrate the functional relevance if compared with the transcriptome data.
6. Why H3K27me3 but not H3K9me3 is detectable in old macronuclei (Figure 2B)? This needs to be explained.

Minor concerns:

1. In Figure 1, its better to show the raw c. p. m. data of scintillation assay, instead of relative activity.
2. Line 259, Fig. 2C should be Fig. 1C.

Reviewer #3 (Remarks to the Author):

The Polycomb-group protein Enhancer of zeste is a component of the ancient PRC2 complex and is well known for its role in directing H3K27me3 methylation and regulation of developmental genes such as homeotic selector genes. In this interesting paper, the authors analyse the role of the *Paramecium tetraurelia* Enhancer of zeste-like protein 1 (Ezl1) gene and show that unusually it is required for H3K9me3 methylation as well as for H3K27me3 and for silencing of many transposable elements. In vitro assays using *Xenopus* histone octamers and immunoprecipitates (IPs) from GFP-tagged EZL1 from transgenic *Paramecium* indicate that EZL1 can methylate H3K9 and H3K27. This is supported by in vivo experiments where EZL1 is knocked down by RNAi resulting in loss of H3K9me3 and H3K27me3 as judged by immunostaining of macronuclei and by ChIP of selected transposons. Transcriptional profiling of Ezl1 RNAi lines suggests that predominantly transposons are deregulated. The authors suggest that the ancestral role of Ez may have been as a dual specificity methyltransferase involved in transposon silencing, and subsequently acquired H3K27 and developmental target specificity.

The findings I think are exciting and of broad interest, and the use of *Paramecium* adds diversity to current studies which are mainly on animals and plants. I am not expert enough on *Paramecium* biology or bioinformatics to judge on the finer details of the transcriptomic analysis or the stages

and genetic strategies used. It does strike me that there are some potential issues with the functional analysis as outlined below. Overall I found the paper interesting and well written and think it is suitable for Nature communications, but it could be strengthened by addressing some of the issues listed below.

1. The in vitro assays in Figure 1 use GFP-Ezl1 partially purified by IP from Paramecium rather than for example EZL1 expressed and purified from Baculovirus or E coli (which may require other components if EZL1 acts in a PRC2 like complex). This is problematic as it is not clear what other Paramecium proteins are in the IP and how they contribute to the activities. The control (IP of catalytically inactive EZL1) is important and adds confidence that the EZL1 activity is needed for H3K9 methylation, however it doesn't eliminate the possibility that there are other Paramecium proteins that co-IP with EZL1 and catalyse H3K9 methylation downstream of H3K27 methylation i.e. that EZL1 specifically methylates H3K27 and something else in the extract provides the H3K9me3 methylation activity. This seems an issue given that Paramecium has three other Ez like proteins that are expressed at the stage used for the IPs (12 hours after onset of sexual events) as well as other SET domain proteins. Also there is background activity in the Ezl1H526A catalytic mutant IP samples which suggests that there are other activities beyond Ezl1 in the extracts. The fact that K27A mutated H3 is methylated more than K9A/K27A H3 or the Ezl1H526A sample perhaps justifies the conclusion that Ezl1 can catalyse H3K9 methylation and that prior H3K27 methylation is not needed for this. However, Figure 1 would be strengthened for example by Western analysis with e.g H3K9me3 antibodies to confirm that this is catalysed, as the current data show that 3H SAM incorporation is reduced by H3K9A/K27A substrates but doesn't indicate whether this involves mono di or tri methylation. It would also add to include a proteomic (mass spectrometry) analysis of the IP samples to give an idea of what else co purifies with EZL1.

2. The in vitro assays in Figure 1 use recombinant Xenopus histone octamers as substrates. This is problematic as Paramecium H3 has some sequence differences around the H3K9 and H3K27 contexts relative to Xenopus or mammals [KQTARK(me3)STAGNK vs KQTARK(me3)STGGKA and TKAARK(me3)TAP vs TKAARK(me3)SAP]. Presumably this is why the authors raised Paramecium specific H3K9me3 and H3K27me3 antibodies for ChIP using peptides bearing the Paramecium sequences. It is therefore possible that the GFP-Ezl1 IP could be specific for H3K27me3 on Paramecium H3. Given the in vivo data based on immunostaining of Ezl1 knockdown supports a role for Ezl1 in H3K9me3 in vivo (although not necessarily direct) I am not necessarily suggesting they need to repeat the assays with recombinant Paramecium histones, but it should be acknowledged/pointed out as a possible issue in the text.

3. Given there have been several recent papers analysing PRC2 crystal structure and function I was expecting a little more analysis of the Ezl1 sequence and how this compares with other Ez proteins and SET domains. The phylogenetics paper by Shaver et al (Epigenomics 5 301) suggests that Paramecium lacks several of the canonical PRC2 components (SuZ12, ESC), but it would be interesting to know if the Ezl1 sequence differs significantly from canonical Ez proteins in the SET domain particularly in residues important for substrate contact in the active site.

Reviewer #4 (Remarks to the Author):

Summary:

One of the characteristic features of ciliates is separating germline and somatic functions in two distinct types of nuclei (nuclear dimorphism), the diploid germline micronucleus (MIC) and the polyploid somatic macronucleus (MAC). During vegetative growth, the diploid MIC remains transcriptionally silent and the polyploid MAC is responsible for gene expression. Meiosis of the germline MIC initiates the sexual reproduction and new MICs and MACs are differentiated from the zygotic nucleus. Development of the MACs from the zygotic nucleus is accompanied by extensive elimination of germline-specific DNA including transposable elements (TE), minisatellites, and internal eliminated sequences. The somatic MAC does not participate in the sexual events and is fragmented and replaced with the newly developed MACs. The mechanisms of the trans-nuclear genome comparison and the zygotic genome rearrangement are not fully understood. Previous studies reported the appearance of specific histone modifications, H3K9me3 and

H3K27me₃, in the developing MACs during the sexual events in unicellular ciliates *Paramecium tetraurelia* and *Tetrahymena thermophila*. H3K9me₃ and H3K27me₃ are generally associated with the repressive transcription and established by distinct protein machineries in plants and animals. In *T. thermophila*, H3K9me₃ and H3K27me₃ were shown to be involved in the programmed DNA elimination during the sexual development. Ezl1p, a homolog of *Drosophila* E(z) protein of the Polycomb complex, was characterized as a histone methyltransferase for H3K27me₃, which subsequently mediates H3K9 methylation leading to heterochromatin formation and DNA elimination (Taverna and Coyne et al. (2002), Liu and Taverna et al. (2007)). Similarly, Ezl1p was also identified in *P. tetraurelia* as a putative histone methyltransferase responsible for H3K9me₃/H3K27me₃ necessary for the programmed DNA elimination. As in *T. thermophila*, these histone marks co-occur with the sexual development of new MACs and are greatly reduced with EZL1 knockdown in *P. tetraurelia* (Lhuillier-Akakpo et al. (2014), Ignarski et al. (2014)). The manuscript by Frapporti et al. is an extension of their previous work characterizing Ezl1p as a histone methyltransferase required for H3K9me₃ and H3K27me₃, necessary for DNA elimination during the sexual development in *P. tetraurelia* (Lhuillier-Akakpo et al. (2014)). In this work, the authors performed in vitro methyltransferase assay to demonstrate that Ezl1p mediates histone H3K9/H3K27 methylation. While the reconstituted mammalian PRC2 complex displayed a strong preference for K27 over K9 as expected, the methyltransferase activities of Ezl1p immunoprecipitates were reduced to the similar level with the K9A and K27A mutations, suggesting that Ezl1p has a methyltransferase activity for both H3K9 and H3K27 in vitro. The authors also performed in vivo genetic complementation assay. While WT EZL1 transgene complemented the H3K9me₃/H3K27me₃, the DNA elimination, and the viability of the sexual progeny in EZL1 RNAi condition, the catalytic mutant transgene didn't, suggesting the catalytic activity of Ezl1p is indispensable for EZL1 function. The authors further investigated the effect of EZL1 knockdown on the transcription level using RNA-seq analysis and showed that a subset of transposable elements (23% of the annotated) is upregulated. Lastly, the authors demonstrated the co-occurrence of H3K9me₃ and H3K27me₃ on the five individual transposable elements by ChIP-qPCR analysis, however, the loss of these histone marks does not correlate with the expression changes of the transposable elements tested. As detailed below, the main claims made in this manuscript are not sufficiently supported by the data. The role of Ezl1p as a putative histone H3 methyltransferase was previously proposed and partially demonstrated, however, the functional understanding of EZL1 has not been extensively expanded in this manuscript. For these reasons, it is difficult to be enthusiastic about this submission.

Comments/suggestions

1. The authors claim that Ezl1p is a dual histone methyltransferase that catalyzes trimethylation of H3K9 and H3K27 in vitro and in vivo. However, both in vitro and in vivo data do not provide direct evidences that Ezl1p catalyzes trimethylation of H3K9 and H3K27. It would be informative to examine the substrate specificity in vitro with the synthetic peptides containing H3K9/H3K27 with the different methylation status. Also, the reduction of Ezl1p methyltransferase activity with the H3K9A and H3K27A mutants is relatively weak (> 2-fold) suggesting that Ezl1p does not have a strong preference. There is a stronger reduction with the double K9A/K27A mutant, the activity is still significantly higher than the background (Fig. 1C). This might indicate the presence of contaminants in the immunoprecipitates or suggest that Ezl1p would have a methyltransferase activity for other lysine residues on H3 with the low specificity. It would be interesting to test Ezl1p methyltransferase activity for the H3K4A mutant as a control. The in vivo genetic complementation assay also demonstrates that the catalytic activity of Ezl1p is required for the H3K9me₃/H3K27me₃ but it does not prove whether Ezl1p acts a dual methyltransferase or the H3K27me₃ catalyzed by Ezl1p subsequently mediates H3K9me₃. Additionally, the authors need to show that the H3 level is not affected by the RNAi and the transgene conditions.
2. With the time course RNA-seq analysis of the EZL1 RNAi sample, the authors observed that a subset of transposable elements (23%) is differentially expressed, mostly upregulated, while the expression levels of tandem repeats and genes are not changed, which allows the authors to claim that EZL1 represses the transposable elements during the sexual development. Although the fraction of genes with differential expression is smaller (0.2% at the early, 5.8% at the late time point), the number of genes whose expression are changed by EZL1 knockdown (2409 genes at later time point) is higher than the number of transposable elements affected (739 TEs), suggesting that the regulation of gene expression by EZL1 during the sexual development needs to

be more thoroughly analyzed. The authors mentioned that the differentially expressed genes are enriched in the developmentally regulated genes, however, further considerations are not found in the manuscript.

3. The authors performed ChIP-qPCR analysis to determine if the upregulated transposable elements are targeted by Ezh1p methyltransferase. They tested five individual transposable elements, three of which are significantly upregulated by EZL1 RNAi and two are not. All the five transposable elements showed enrichment for H3K9me3 and H3K27me3. This observation proves the co-occurrence of H3K9me3 and H3K27me3 on the same five transposable elements as the authors claimed, however, is not consistent with the authors' proposal that the upregulation of transposable elements is due to the derepression caused by loss of H3K9me3/H3K27me3 in EZL1 RNAi condition. The authors do not provide any consideration with this matter. More transposable elements need to be investigated and, ideally, genome-wide ChIP-seq analysis would be desirable to make a better conclusion on the correlation between the H3K9me3/H3K27me3 enrichment and the differential gene expression during the sexual development.

Response to Referees

Reviewer #1:

Related to the experiments shown in Figure 1:

1) Is the sequence of histone H3 (especially its N-terminal tail) of *Paramecium* identical to that of *Xenopus* and human? If not, some discussion is necessary to justify that the assays using *Paramecium* Ezh1 with vertebrate histones/nucleosomes are valid.

The reviewer is right to point out that the *Paramecium* histone H3 sequence is not identical to that of *Xenopus* used for the *in vitro* HMT assays on recombinant histone octamers or nucleosomes. We chose to use *Xenopus* in these assays for the sake of comparison to mammalian PRC2. Importantly, since the sequence divergence affects more the residues flanking H3K27 than H3K9, we do not expect that it would impact PRC2-Ezh1 specificity toward H3K9. It is worth noting that because of the sequence divergence we generated a specific antibody toward *Paramecium* H3K27me3 for immunostaining and ChIP experiments.

2) The differences between lanes 7/8 and 9 of Figure 1C might not be convincingly large. Are these statistically significant?

We now include a quantification of *in vitro* methyltransferase activities for 3 biological replicates (now Figure 2C). The quantification confirms that the methyltransferase activity of GFP-Ezh1 immunoprecipitates is partially reduced on K9A and K27A, compared to wild type octamers, and that a stronger reduction is observed on double K9A/K27A mutant octamers.

3) Some activity was detected in the lane 9 of Figure 1C. This might suggest that Ezh1 has a methyltransferase activity for some residues other than K9 and K27 of H3. I believe some discussion for this point is necessary in the text.

The reviewer is correct in noting a residual activity on the double K9A/K27A mutant octamers. As is now noted more explicitly in the main text, the residual activity of the Ezh1 immunoprecipitates indicates either that Ezh1 has a methyltransferase activity for residues other than K9 and K27 of H3 and/or that there are contaminants in our pull-downs.

Related to the experiments shown in Figure 2:

4) The maternal MAC contains H3K27me3 in the control cells and it is dependent on the catalytic activity of Ezh1. Curiously, H3K9me3 is not in the maternal MAC even in the control cells. These suggest that Ezh1 catalyzes only H3K27me3 in the maternal MAC but both H3K9me3 and H3K27me3 in the new MAC. First, I believe this point should be discussed in the text.

We agree with the reviewer that Ezh1 activity appears to be regulated during development. Our observations suggest that the Ezh1 complex first catalyzes only H3K27me3 in the maternal MAC, while at a later stage of development it catalyzes both H3K9me3 and H3K27me3 in the new MAC. We now discuss this point in the first section of the Discussion.

Second, I think it is important to purify EGFP-Ezh1 complex from cells before the new MAC development and perform a methyltransferase assay like in Figure 1 to ask if it shows preference for H3K27.

We agree that it would be very interesting to compare the HMT activity of Ezh1 complexes isolated when it is in the new developing MAC and both K27me3 and K9me3 are detected (our experiments) to that of Ezh1 complexes when it is in the maternal MAC and only K27me3 is detected. Unfortunately, we cannot synchronize cells during development and the GFP-Ezh1 protein is found in the fragments of the maternal MAC in a transient manner (Lhuillier-Akakpo et al., 2014). This experiment is thus not technically feasible.

Related to the experiments shown in Figure 4:

5) Are the PCR primers for ChIP-qPCR designed to specifically amplify a particular copy of a transposon family or to amplify a whole transposon family? If the latter is the case, it would not be possible to conclude that H3K9me3 and H3K27me3 co-occur on a same copy of a transposon.

Indeed, we designed PCR primers to amplify a particular copy of a transposon family, which is possible because different TE copies within a family are divergent enough in their nucleotide sequence. We have now expanded our ChIP-qPCR analysis to more TE copies. We included two pairs of primers, each one designed to specifically amplify one individual copy of the same transposon family, and sequenced the resulting PCR products (see Methods). As expected, each pair of primers only amplifies a single TE copy, validating that our design strategy distinguishes TE copies within a family.

Related to the last section of Discussions:

6) Does *Paramecium* have homologs of PRC2 subunits beside Ezh1? If yes, it would be interesting to discuss those proteins as well.

No homolog of Eed or Suz12 could be identified by sequence homology search, likely because of sequence

divergence. Our preliminary results suggest that Ezl1 indeed belongs to a multi-subunit complex. The composition of the Ezl1 core complex appears to have both similarities and differences with that of the mammalian PRC2. Further investigations are therefore needed before reporting these findings. The extensive characterization of Ezl1 complex subunits will be the subject of future studies.

Reviewer #2:

Major concerns:

1. It is not totally surprising that an enzyme can do both H3K9 and H3K27 methylation. As the authors cited, PRC2 is known to display minor activities at H3K9. Moreover, G9a and GLP, the two major H3K9 dimethylase in mammals are also known to display some activities at H3K27. Thus, the potential dual substrate-specificity of Ezl1 does not appear to be novel enough.

We agree with this reviewer that PRC2 displays minor activity toward K9 *in vitro*, which our results confirm (new Fig. 2). However, to our knowledge, no *in vivo* activity toward K9 has been convincingly documented for PRC2 in either flies or mammals, in marked contrast to what we observe for *Paramecium* Ezl1. Conversely, we are not aware of reports showing that G9A directly catalyzes H3K27me3 deposition *in vivo*. Therefore, we would argue that the dual substrate-specificity of Ezl1 is in fact without precedent from an *in vivo* functional standpoint.

2. The quality of purified Ezl1 is not satisfying. As the authors acknowledge, they cannot rule out the contribution of contaminant proteins in their *in vitro* activity assays. I agree that this is a valid concern. First of all, there is not silver staining of the purified protein or protein complex to demonstrate the purity of their preparation.

We performed our HMT assays with GFP pull-downs. We don't have enough material to further purify it by chromatography; however, we used a very stringent wash buffer (up to 600mM salt) to limit the presence of contaminants. Given that we do not know the precise composition of the Ezl1 complex, we cannot perform *in vitro* HMT assays with a reconstituted complex but hope to do so in the future. We reasoned that the catalytically dead mutant was very important to control for possible contaminants for the *in vitro* HMT assays and included the Ezl1^{H526A} mutant for that reason.

Secondly, their catalytic dead mutant is not dead (Fig. 1A) and their enzyme remain active at K9/K27 double mutant substrate (Fig. 1C). This certainly raise a possibility that some of the HMT activities observed in their *in vitro* assay come from contaminant proteins.

The reviewer is correct in noting that despite our best efforts there is some residual activity with the Ezl1^{H526A} mutant IP on wild type octamers, as well as with the wild type Ezl1 IP on the double K9A/K27A mutant octamers. The fact that these two residual activities are similar led us to conclude that they represent the background inherent to the approach. Since this activity is absent in GFP pull-downs performed from non-transformed cells (new Supplementary Fig. 2), it suggests that it is not bound to the beads but rather pulled down with GFP-Ezl1.

3. Enhancer of Zeste and its homologues proteins from mammals, fly, *C. elegans* and even thermophilic fungi exist in multi-subunit complex and do not exert HMT activity alone. Does Ezl1 from *Paramecium tetraurelia* also forms a complex with other proteins? This should be answered since the authors have already purified it from the cells.

Our preliminary results suggest that Ezl1 indeed belongs to a multi-subunit complex. While the apparent composition of the Ezl1 complex presents similarity to that of the mammalian PRC2, we also identified additional polypeptides which we need to confirm before publishing them.

4. Does *Paramecium tetraurelia* have other enzymes responsible for H3K9me3 and H3K27me3?

Knock-down of the *EZL1* gene abolishes H3K9me3 and H3K27me3 signals in immunofluorescence experiments (Fig. 3). Therefore, it does appear that the contribution of any potentially redundant enzymes is limited in this developmental context.

5. What is the extent of colocalization of H3K9me3 and H3K27me3 at the genome-wide level? ChIP-seq experiments will be helpful to address this and also to demonstrate the functional relevance if compared with the transcriptome data.

We agree that examining the extent of the co-occurrence of H3K9 and H3K27 genome-wide will be interesting but is beyond the scope of the present study. Nonetheless, we have expanded our ChIP-qPCR analysis: Fig. 5 now includes 7 additional TEs and 4 additional genes.

6. Why H3K27me3 but not H3K9me3 is detectable in old macronuclei (Figure 2B)? This needs to be explained.

The reviewer is correct in noting that only H3K27me3 is detected in the maternal MAC, while both H3K9me3 and H3K27me3 are detected in the new MAC at later stages of development. This is entirely consistent with our observations that GFP-Ezl1 first localizes in the maternal MAC at early stages of development then exclusively localizes in the new MAC (Lhuillier-Akakpo et al., 2014). We have clarified this point in the legend of Fig. 3. In both nuclei, deposition of the histone marks requires Ezl1 catalytic activity (Fig. 3). These observations suggest that the Ezl1 complex catalyzes only H3K27me3 in the maternal MAC, while it catalyzes both H3K9me3 and H3K27me3 in

the new MAC. We now mention in the discussion that Ezl1 activity appears to be regulated during development.

Minor concerns:

1. In Figure 1, it's better to show the raw c. p. m. data of scintillation assay, instead of relative activity. We tried to use c.p.m. instead of relative activity but this turned out to be difficult. Indeed, while the relative activity of Ezl1 is very consistent from one experiment to another (normalized to the activity on wild type substrate), the absolute numbers differ. We should emphasize that to have true biological replicates, our IPs were done on distinct extracts. This approach implies different levels of expression of the transgene and therefore variable amounts of Ezl1 in the pull-downs, which causes variability in the raw c.p.m. data.
2. Line 259, Fig. 2C should be Fig. 1C. Figure numbers have changed.

Reviewer #3:

1. The *in vitro* assays in Figure 1 use GFP-Ezl1 partially purified by IP from *Paramecium* rather than for example EZL1 expressed and purified from Baculovirus or *E. coli* (which may require other components if EZL1 acts in a PRC2 like complex). This is problematic as it is not clear what other *Paramecium* proteins are in the IP and how they contribute to the activities. The control (IP of catalytically inactive EZL1) is important and adds confidence that the EZL1 activity is needed for H3K9 methylation, however it doesn't eliminate the possibility that there are other *Paramecium* proteins that co-IP with EZL1 and catalyse H3K9 methylation downstream of H3K27 methylation i.e. that EZL1 specifically methylates H3K27 and something else in the extract provides the H3K9me3 methylation activity. While we cannot formally rule out the reviewer's concern, our new result showing that recombinant octamers and octamers chemically trimethylated on H3K27 are methylated by Ezl1 to a comparable extent strongly supports a dual substrate specificity for Ezl1 (new Fig. 2D).

This seems an issue given that *Paramecium* has three other Ez like proteins that are expressed at the stage used for the IPs (12 hours after onset of sexual events) as well as other SET domain proteins. Our data indicate that Ezl1 is essential *in vivo* for H3K9me3 and H3K27me3 deposition. Knock-down of the *EZL1* gene abolishes H3K9me3 and H3K27me3 signals in immunofluorescence (Fig. 3). Therefore, enzymes potentially redundant with Ezl1 cannot compensate for its deletion at this stage of development.

Also there is background activity in the Ezl1H526A catalytic mutant IP samples which suggests that there are other activities beyond Ezl1 in the extracts. Indeed, despite our efforts (new Fig. 2), there is some residual activity in the Ezl1 H526A mutant IP. This activity is comparable to that of the wild type Ezl1 IP on the K9A/K27A double mutant octamers, consistent with the hypothesis that Ezl1 activity is directed toward lysines 9 and 27 of histone H3.

The fact that K27A mutated H3 is methylated more than K9A/K27A H3 or the Ezl1H526A sample perhaps justifies the conclusion that Ezl1 can catalyse H3K9 methylation and that prior H3K27 methylation is not needed for this. We agree with the reviewer (see below).

However, Figure 1 would be strengthened for example by Western analysis with e.g. H3K9me3 antibodies to confirm that this is catalysed, as the current data show that 3H SAM incorporation is reduced by H3K9A/K27A substrates but doesn't indicate whether this involves mono di or tri methylation. We agree with the reviewer that this would strengthen our conclusions. We tried to perform the suggested experiment, but in our experimental conditions we could not detect any robust signal (either for H3K27 or for H3K9 methylation). Nonetheless, we believe that our *in vivo* complementation assay showing that the catalytic domain of Ezl1 is required to rescue K27me3 and K9me3 argues in favor of the idea that Ezl1 generates these modifications.

It would also add to include a proteomic (mass spectrometry) analysis of the IP samples to give an idea of what else co-purifies with EZL1. Our preliminary data indicate that Ezl1 belongs to a multi-subunit complex whose characterization will be investigated in the future.

2. The *in vitro* assays in Figure 1 use recombinant *Xenopus* histone octamers as substrates. This is problematic as *Paramecium* H3 has some sequence differences around the H3K9 and H3K27 contexts relative to *Xenopus* or mammals [KQTARK(me3)STAGNK vs KQTARK(me3)STGGKA and TKAARK(me3)TAP vs TKAARK(me3)SAP]. Presumably this is why the authors raised *Paramecium* specific H3K9me3 and H3K27me3 antibodies for ChIP

using peptides bearing the Paramecium sequences. It is therefore possible that the GFP-Ez11 IP could be specific for H3K27me3 on Paramecium H3. Given the *in vivo* data based on immunostaining of Ez11 knockdown supports a role for Ez11 in H3K9me3 *in vivo* (although not necessarily direct) I am not necessarily suggesting they need to repeat the assays with recombinant Paramecium histones, but it should be acknowledged/pointed out as a possible issue in the text.

The reviewer is correct in pointing out the sequence disparities in histone H3 between *Paramecium* and *Xenopus*. We now discuss this in the main text and explain that we chose to use *Xenopus* histones in order to directly compare the activity of the GFP-Ez11 IP to that of mammalian PRC2.

3. Given there have been several recent papers analysing PRC2 crystal structure and function I was expecting a little more analysis of the Ez11 sequence and how this compares with other Ez proteins and SET domains. The phylogenetics paper by Shaver et al (Epigenomics 5 301) suggests that Paramecium lacks several of the canonical PRC2 components (SuZ12, ESC), but it would be interesting to know if the Ez11 sequence differs significantly from canonical Ez proteins in the SET domain particularly in residues important for substrate contact in the active site.

We are grateful to this reviewer for pushing us to perform this analysis as we feel it will greatly strengthen the impact of the article. We now include a comparative homology modeling of the the SET domain of Ez11 with Ezh2 in two additional figures (new Fig. 1 and Supplementary Fig. 1).

Reviewer #4:

Comments/suggestions

1. The authors claim that Ez11p is a dual histone methyltransferase that catalyzes trimethylation of H3K9 and H3K27 *in vitro* and *in vivo*. However, both *in vitro* and *in vivo* data do not provide direct evidences that Ez11p catalyzes trimethylation of H3K9 and H3K27. It would be informative to examine the substrate specificity *in vitro* with the synthetic peptides containing H3K9/H3K27 with the different methylation status.

We agree with the reviewer that our *in vitro* assays alone do not make it possible to claim that Ez11 catalyzes trimethylation and similarly that while our *in vivo* assays show that Ez11 is required for H3K27me3 and H3K9me3, this does not demonstrate that these marks are directly catalyzed by Ez11. Nevertheless, both results considered together strongly argue in favor of this hypothesis.

While interesting in principle, the experiment proposed by the reviewer presents considerable technical difficulties. Indeed, peptides are very poor substrates for PRC2. This experiment would therefore require much more enzyme than what we can obtain in our IPs. To bypass this limitation, we tried instead to use chemically modified histones that mimic the different levels of methylation of H3K27. However, Ez11 methylates unmodified chromatin comparably to chromatin already fully methylated on H3K27 (probably due to methylation of H3K9), which prevents us from using this approach to determine whether Ez11 can still methylate H3K27me2, for instance (see new Fig. 2D).

Also, the reduction of Ez11p methyltransferase activity with the H3K9A and H3K27A mutants is relatively weak (> 2-fold) suggesting that Ez11p does not have a strong preference. There is a stronger reduction with the double K9A/K27A mutant, the activity is still significantly higher than the background (Fig. 1C). This might indicate the presence of contaminants in the immunoprecipitates or suggest that Ez11p would have a methyltransferase activity for other lysine residues on H3 with the low specificity. It would be interesting to test Ez11p methyltransferase activity for the H3K4A mutant as a control.

The reviewer raises two alternative possibilities that could account for the difference in activity between lanes 9 and 10 of Fig. 1C (now new Fig. 2C). First, a contaminant activity could be present in our IPs. We agree that, despite our best efforts, this is most likely indeed the case. Importantly, changes in chromatin structure resulting from *EZL1* knockdown could lead to differential recovery of this activity in the two IPs, and it could thus be sufficient to account for the signal remaining both with the H3-K9A/K27A double mutant and with the catalytically dead mutant. Alternatively, as also mentioned by the reviewer, the activity of wild type Ez11 on the K9A/K27A double mutant might be higher than the catalytically dead background because of a weak activity of Ez11 on another histone H3 residue. We have performed HMT assays on H3K4A mutant octamers as suggested and, consistent with either of the two proposed explanations, the activity is slightly reduced by this mutation (see panel below).

The *in vivo* genetic complementation assay also demonstrates that the catalytic activity of Ezl1p is required for the H3K9me3/H3K27me3 but it does not prove whether Ezl1p acts a dual methyltransferase or the H3K27me3 catalyzed by Ezl1p subsequently mediates H3K9me3.

As stated above, we agree that the *in vivo* complementation assay does not formally demonstrate that Ezl1 catalyzes H3K27me3 and H3K9me3 directly. Nonetheless, the observation that Ezl1 has less activity on H3-K9A/K27A double mutant octamers than on the K27A single mutant suggests that Ezl1 directly methylates both K9 and K27.

To further corroborate our hypothesis, we investigated whether Ezl1 can still methylate H3 on a premethylated H3K27me3 substrate. This experiment revealed that Ezl1 maintains a robust activity when H3K27 is already fully methylated, thus providing additional support for a dual specificity (new Fig. 2D).

Additionally, the authors need to show that the H3 level is not affected by the RNAi and the transgene conditions. To address this question, we performed Western blot analyses of total protein extracts in uninjected cells and in cells transformed with *GFP-EZL1* and *GFP-EZL1^{H526A}* transgenes upon control RNAi and *EZL1* RNAi. The results with GFP, H3 and alpha-tubulin antibodies (for normalization) are shown below. H3 protein levels are similar in all conditions.

2. With the time course RNA-seq analysis of the *EZL1* RNAi sample, the authors observed that a subset of transposable elements (23%) is differentially expressed, mostly upregulated, while the expression levels of tandem repeats and genes are not changed, which allows the authors to claim that *EZL1* represses the transposable elements during the sexual development. Although the fraction of genes with differential expression is smaller (0.2% at the early, 5.8% at the late time point), the number of genes whose expression are changed by *EZL1* knockdown (2409 genes at later time point) is higher than the number of transposable elements affected (739 TEs), suggesting that the regulation of gene expression by *EZL1* during the sexual development needs to be more thoroughly analyzed. The authors mentioned that the differentially expressed genes are enriched in the developmentally regulated genes, however, further considerations are not found in the manuscript.

The referee is correct in noting that the number of TE copies is quite modest. First of all, the data on TE expression were obtained using the presently available set of germline TEs, based on manual curation (Guerin et al., 2017). We do not have a more comprehensive picture of TE landscape in *Paramecium* because we do not yet have a complete, high quality germline genome assembly. In other words, we have probably underestimated the true number of TEs in the *Paramecium* germline as well as the number of TEs that are transcribed upon *EZL1* RNAi. Furthermore, it is important to keep in mind, that the evolutionary history of whole genome duplications has led to the unusually high number of close (often functionally redundant) paralogs in *Paramecium*. That is why the inferences we have drawn are based on percentages rather than raw numbers.

The dramatic impact of *EZL1* KD on TE expression and the modest effect on gene expression can also be appreciated by comparing the volcano plots in Figs. 3-4 and Supplementary Fig. 4: the effects of *EZL1* KD on TE expression leads to much greater fold-changes (282.52-fold change considering all TE differentially expressed) than the effects on gene expression (4.83-fold change considering all up-regulated differentially expressed genes). To explain this analysis, we added a Supplementary Table, Supplementary Table 3.

Concerning the differentially expressed genes, we now provide Supplementary Data 1 to list all differentially expressed genes, and a new Supplementary Table to present the enrichment in development genes i.e. genes differentially expressed during sexual processes (Supplementary Table 2). The most enriched category is that of genes specifically expressed at the time of genome rearrangements ("Intermediate peak" genes).

We chose not to present in detail the effects of *EZL1* KD on gene expression in the main text. First, because the effects on TE de-repression are more striking. Second, though our data does suggest that PRC2 in ciliates could be involved in regulation of genes required for development of the new macronucleus during the sexual cycle, the effects of *EZL1* KD on gene expression may arise from the retention of about 60% of all IESs after *EZL1* KD (Lhuillier-Akakpo et al., 2014). Further work will be required to tease apart direct effects of Ezl1 on gene expression, indirect effects, and effects owing to IES retention.

3. The authors performed ChIP-qPCR analysis to determine if the upregulated transposable elements are targeted

by Ezl1p methyltransferase. They tested five individual transposable elements, three of which are significantly upregulated by EZL1 RNAi and two are not. All the five transposable elements showed enrichment for H3K9me3 and H3K27me3. This observation proves the co-occurrence of H3K9me3 and H3K27me3 on the same five transposable elements as the authors claimed, however, is not consistent with the authors' proposal that the upregulation of transposable elements is due to the derepression caused by loss of H3K9me3/H3K27me3 in EZL1 RNAi condition. The authors do not provide any consideration with this matter.

Our ChIP data indicate that all individual TE copies tested showed enriched occupancy of H3K27me3 and H3K9me3, regardless of whether these copies were transcriptionally upregulated upon *EZL1* RNAi and our RNAseq data indicate that 24 % of annotated TE copies are upregulated upon *EZL1* RNAi. We think that our data support the conclusion that loss of H3K27me3 and H3K9me3 upon *EZL1* RNAi coincides with TE transcriptional derepression. However, releasing transcriptional repression does not necessarily mean that the genes will be transcribed. Indeed, an activating input is also required for gene activation. We have tried to identify distinguishing features among the subset of TEs that get upregulated however we did not find anything specific. For instance, no correlation could be observed between the age of the copy and its expression level upon *EZL1* RNAi, indicating that other, unidentified factors may explain this difference. We have tried to make this clearer in the discussion section.

More transposable elements need to be investigated and, ideally, genome-wide ChIP-seq analysis would be desirable to make a better conclusion on the correlation between the H3K9me3/H3K27me3 enrichment and the differential gene expression during the sexual development.

We have now expanded our ChIP qPCR analysis to more TE copies and more genes. The results are shown in new Fig. 5 and confirmed our previous observations: all TE copies are specifically enriched in H3K9me3 and H3K27me3 in an Ezl1-dependent manner.

Reviewers' comments:

Reviewer #1 (Remarks to the Author):

I think the authors have addressed all my comments raised in the previous round of review and I have no additional comments for the revised manuscript.

Reviewer #2 (Remarks to the Author):

The authors addressed my minor concerns but it is sad that they are very reluctant in addressing my main concerns experimentally.

1. The quality of purified Ezl1 is not satisfying. The author argue that their Ezl1 was purified under highly stringent conditions to limit the presence of contaminants. This is great, but there is no proof!!! Why a silver staining of the purified material cannot be shown as requested? Because it was too dirty? The authors state that they do not know the precise composition of the Ezl1 complex. Why? Under such stringent purification condition that the authors claim, they should be able to purify the complex and identify the subunits of the complex. The authors argue that a mutant complex is a good control, which I agree, but not without showing the silver staining of the purified wild-type and mutant complex side-by-side! Moreover, as I mentioned previously, there dead mutant is in fact active (new Fig. 2A) and there wild-type enzyme is active on substrates with both K9 and K27 mutated (new Fig. 2C). This is a great indication that they have contaminant activities in their purified material! Unfortunately, the authors did nothing but arguing.
2. I requested ChIP-seq and RNA-seq experiments and the authors argued that they are beyond the scope of this manuscript, which I disagree. Nature Communication is a respected journal and functional relevance of their biochemical activity is required. Moreover, this experiment will tell us to what extent Ezl1 contributes to H3K9me3 and H3K27me3 in the genome and whether such contribution has functional relevance.

Reviewer #3 (Remarks to the Author):

The authors have made a sincere attempt to address the comments raised by the referees and I believe the paper has been considerably improved. The structural modelling data adds to the data that Ezl1 may have dual specificity. Some of the issues that have been raised, for example the residual activity present in the extracts from Ezl1 catalytically dead mutants, it has not been possible to address for technical reasons, but the authors do now point these issues out in the text, which is welcome. I don't think the possibility that H3K9me3 is indirect (i.e. EZL1 is required for H3K27me3 and this is needed for a different enzyme to add H3K9me3) is wholly eliminated. However, I think that collectively their data makes a strong case for dual specificity of EzL1 and an interesting paper of broad biological interest for the field. The role of an Ez homologue in transposon silencing I also think is of broad interest. I think the paper should be accepted and is very suitable for Nature Communications.

Reviewer #4 (Remarks to the Author):

In this revised manuscript, the authors performed comparative homology modeling of the catalytic SET domain of Ezl1 and the human Ezh2 to support that Ezl1 is a dual methyltransferase specific for H3K27 and H3K9. While Ezh2 and the modeled Ezl1 are structurally conserved in the catalytic SET domain, the H3 substrate-binding site of the modeled Ezl1 is potentially similar to that of H3K9 methyltransferase, suggesting that Ezl1 might have a substrate preference for both H3K27 and H3K9. However, it is the prediction based on the modeled structure of Ezl1 and there is no experimental evidence to support this idea. A mutational analysis needs to be accompanied to understand the role of the residues proposed to be involved in H3K9 substrate binding. One major issue is that the GFP-Ezl1 immunoprecipitates do not have a strong histone

methyltransferase activity for H3K9 in vitro. In the revised Fig. 2C including the biological replicates, it is more apparent that Ezi1 has a stronger preference for H3K27. The reduction of Ezi1 HMT activity measured in K9A is rather modest and, in fact, no significant difference is observed when compared to PRC2 (Fig. 2C lanes 1-2, 6-7). The authors additionally included Fig. 2D showing that Ezi1 methylates another target(s) on H3 other than K27 to a lower extent. However, this does not necessarily mean that the residual activity is solely due to H3K9 methylation. Considering that the effect of K9A on Ezi1 HMT activity is mild and also Ezi1 still has a residual activity in K9A/K27A double mutant possibly due to the contaminants, the results of in vitro HMT assay with the Ezi1 immunoprecipitates are not satisfyingly convincing.

In Fig. 3B, the H3K9me3 is detected only in the developing MACs while H3K27me3 is observed in both the fragments from the old vegetative MACs and the developing MACs. In discussion, the authors claim that Ezi1 activity is developmentally regulated so it catalyzes only H3K27me3 in the maternal MAC but catalyzes both H3K27me3 and H3K9me3 in the new MAC. This is a reasonable interpretation but it is rather descriptive without providing any supporting evidence or giving a similar example. In addition, Ignarski et al. previously observed both H3K9me3 and H3K27me3 in the fragmented parental MAC and the developing MAC of *Paramecium tetraurelia*.

Although I agree that Ezi1 mediates H3K27me3 and H3K9me3 (directly or indirectly) in *P. tetraurelia* based on the in vivo data, there are some major issues to be addressed as detailed above. I would not recommend this manuscript for publication.

Reviewers' comments:

Reviewer #1 (Remarks to the Author):

I think the authors have addressed all my comments raised in the previous round of review and I have no additional comments for the revised manuscript.

Reviewer #2 (Remarks to the Author):

The authors addressed my minor concerns but it is sad that they are very reluctant in addressing my main concerns experimentally.

1. The quality of purified Ezl1 is not satisfying. The author argue that their Ezl1 was purified under highly stringent conditions to limit the presence of contaminants. This is great, but there is no proof!!! Why a silver staining of the purified material cannot be shown as requested? Because it was too dirty? The authors state that they do not know the precise composition of the Ezl1 complex. Why? Under such stringent purification condition that the authors claim, they should be able to purify the complex and identify the subunits of the complex. The authors argue that a mutant complex is a good control, which I agree, but not without showing the silver staining of the purified wild-type and mutant complex side-by-side! Moreover, as I mentioned previously, there dead mutant is in fact active (new Fig. 2A) and there wild-type enzyme is active on substrates with both K9 and K27 mutated (new Fig. 2C). This is a great indication that they have contaminant activities in their purified material! Unfortunately, the authors did nothing but arguing.

To address these concerns, we have performed new *in vitro* experiments using an array of additional conditions and more stringent immunopurification methodologies, while at the same time demonstrating that our preparations are highly pure and consistent between replicates. Our new results robustly corroborate our previous conclusions, and make the case for our model much more compelling. In detail, our manuscript now contains the following additions:

1. We have further purified the Ezl1 complex from *Paramecium* nuclear extracts through a double affinity procedure. The composition of the wild-type and mutant Ezl1 complexes were analyzed side by side on silver-stained gels. This analysis showed that the two preparations are very much alike (Supplementary Figure 2 panel C).
2. We performed histone methyltransferase (HMT) assays with the double-affinity purified Ezl1 complexes and confirmed the results we had previously obtained with GFP-Trap purification: the HMT activity towards H3 is greatly diminished with the H3 K9/K27 double mutant (Supplementary Figure 2 panel D). We further validated this result using *Paramecium* histone H3 as a substrate. Recapitulating our previous results, reduced methyltransferase activity was detected on K9/K27 double-mutant H3 (Figure 2 panels D and E; and Supplementary Figure 2 panel D).
3. We used mass spectrometry to determine the sites of methylation in the *in vitro* HMT assays using the double-affinity purified Ezl1 complexes. We now demonstrate that both K9 and K27 of *Paramecium* H3 are methylated in an Ezl1-dependent manner (Figure 2 panel E, Supplementary Table 1).
4. Mass spectrometry analysis also indicated that the residual methylation detected using the K9/K27 *Paramecium* H3 mutant can be attributed to a weak activity of Ezl1 toward an additional lysine, an effect that is much more pronounced when positions 9 and 27 are mutated and can no longer be methylated (Supplementary Table 1).

The mass spectrometry proteomics data have been deposited to the ProteomeXchange Consortium via the PRIDE partner repository with the data set identifier PXD012170.

Project DOI: 10.6019/PXD012170

Reviewer account details:

Username: reviewer00685@ebi.ac.uk

Password: xxOHCyeZ

These additions are included in the following new Figures:

- Figure 2, panels D-E
- Supplementary Figure 2, panels C-D-E
- Supplementary Table 1

Please note that Figure 2 ex-panel D has been removed. It was no longer needed because of the mass spectrometry data.

Finally, the reviewer is correct to point out that the Ezl1^{H526A} mutant enzyme is not entirely devoid of methyltransferase activity. We now describe the activity of this point mutant more accurately as “severely compromised.”

2. I requested ChIP-seq and RNA-seq experiments and the authors argued that they are beyond the scope of this manuscript, which I disagree. Nature Communication is a respected journal and functional relevance of their biochemical activity is required. Moreover, this experiment will tell us to what extent Ezl1 contributes to H3K9me3 and H3K27me3 in the genome and whether such contribution has functional relevance.

As suggested by the editor for the first round of revision, we had extended our ChIP-qPCR analysis to more genomic loci. We had also extended our RNA-seq data analyses. Notably, we had performed a more thorough analysis of the regulation of gene expression by *EZL1* during development.

We have now successfully performed ChIP-seq in *Paramecium* and include our unbiased genome-wide analysis, which confirms and extends our candidate-based results obtained by ChIP-qPCR and reveals that H3K9me3 and H3K27me3 both decorate transposable elements (new Figure 5 panel C, new Supplementary Figure 5, new Supplementary Figure 6).

These additions are presented in the following new Figures:

- Figure 5, panel C
- Supplementary Figure 5
- Supplementary Figure 6
- Supplementary Table 6

Reviewer #3 (Remarks to the Author):

The authors have made a sincere attempt to address the comments raised by the referees and I believe the paper has been considerably improved. The structural modelling data adds to the data that Ezl1 may have dual specificity. Some of the issues that have been raised, for example the residual activity present in the extracts from Ezl1 catalytically dead mutants, it has not been possible to address for technical reasons, but the authors do now point these issues out in the text, which is welcome. I don't think the possibility that H3K9me3 is indirect (i.e. EZL1 is required for H3K27me3 and this is needed for a different enzyme to add H3K9me3) is wholly eliminated. However, I think that collectively their data makes a strong case for dual specificity of EzL1 and an interesting paper of broad biological interest for the field. The role of an Ez homologue in transposon silencing I also think is of broad interest. I think the paper should be accepted and is very suitable for Nature Communications.

The HMT assays performed with more stringent immunopurification methodologies (new Figure 2 panels D-E, new Supplementary Figure 2 panels C-D, new Supplementary Table 1) confirmed the results we had previously obtained with GFP-Trap purification and thus, provide stronger support to the idea that Ez1 directly methylates K9 as well as K27. Our mass spectrometry analysis demonstrates that both K9 and K27 of *Paramecium* H3 are methylated in an Ezl1-dependent manner (new Figure 2 new panel E, new Supplementary Table 1).

Reviewer #4 (Remarks to the Author):

In this revised manuscript, the authors performed comparative homology modeling of the catalytic SET domain of Ezl1 and the human Ezh2 to support that Ezl1 is a dual methyltransferase specific for H3K27 and H3K9. While Ezh2 and the modeled Ezl1 are structurally conserved in the catalytic SET domain, the H3 substrate-binding site of the modeled Ezl1 is potentially similar to that of H3K9 methyltransferase, suggesting that Ezl1 might have a substrate preference for both H3K27 and H3K9. However, it is the prediction based on the modeled structure of Ezl1 and there is no experimental evidence to support this idea. A mutational analysis needs to be accompanied to understand the role of the residues proposed to be involved in H3K9 substrate binding.

As suggested by reviewer #3, we had performed for the first round of revision a comparative homology modeling of the the SET domain of Ezl1 with Ezh2 (Figure 1). This analysis showed that the most significant

differences occur in the I-SET region that contributes to the substrate binding site, suggesting that Ezl1 may have different substrate preferences from Ezh2. We agree with this reviewer that it would be very interesting to test the role of specific residues. We have generated mutants in which chosen residues were substituted in Ezl1 by their Ezh2 counterparts and tested them using an *in vivo* genetic complementation approach. Unfortunately, we have so far been unable to uncouple methylation on K27 and on K9.

One major issue is that the GFP-Ezl1 immunoprecipitates do not have a strong histone methyltransferase activity for H3K9 *in vitro*. In the revised Fig. 2C including the biological replicates, it is more apparent that Ezl1 has a stronger preference for H3K27. The reduction of Ezl1 HMT activity measured in K9A is rather modest and, in fact, no significant difference is observed when compared to PRC2 (Fig. 2C lanes 1-2, 6-7). The authors additionally included Fig. 2D showing that Ezl1 methylates another target(s) on H3 other than K27 to a lower extent. However, this does not necessarily mean that the residual activity is solely due to H3K9 methylation. Considering that the effect of K9A on Ezl1 HMT activity is mild and also Ezl1 still has a residual activity in K9A/K27A double mutant possibly due to the contaminants, the results of *in vitro* HMT assay with the Ezl1 immunoprecipitates are not satisfyingly convincing.

see response to reviewer #2, point 1.

In Fig. 3B, the H3K9me3 is detected only in the developing MACs while H3K27me3 is observed in both the fragments from the old vegetative MACs and the developing MACs. In discussion, the authors claim that Ezl1 activity is developmentally regulated so it catalyzes only H3K27me3 in the maternal MAC but catalyzes both H3K27me3 and H3K9me3 in the new MAC. This is a reasonable interpretation but it is rather descriptive without providing any supporting evidence or giving a similar example. In addition, Ignarski et al. previously observed both H3K9me3 and H3K27me3 in the fragmented parental MAC and the developing MAC of *Paramecium tetraurelia*.

At the reviewer's request, we had included in the revised version a discussion about the possibility that Ezl1 activity is developmentally regulated, based on our observation that i) Ezl1 is found first in the maternal MAC then in the new MAC and that ii) Ezl1 catalyzes only K27me3 in the maternal MAC and both K9me3 and K27me3 in the new MAC. These observations have been made using two independent polyclonal antibodies whose specificity was validated by dot blot and competition assays (Lhuillier-Akakpo et al 2014 and this study). In the cited paper (Ignarski et al 2014), the same commercial antibody as in Lhuillier-Akakpo et al 2014 was used. In contrast to our results, the authors did detect K9me3 in the maternal MAC. This discrepancy may be explained by technical differences. It is worth pointing out that the Ignarski paper did not validate the specificity of the polyclonal antibody batch they used. Of note, the same lab recently published a study (Bhullar et al 2018 NAR) in which the same commercial K9me3 antibody was again used, and this time, the authors no longer detect H3K9me3 in the maternal MAC. The authors only detect K9me3 in the new MAC, as we do.

Because we agree it is highly speculative, we decided to remove from the discussion the hypothesis that Ezl1 is developmentally regulated. Instead, we now mention in the legend of Figure 3 that H3K9me3 is detected only in the developing MACs while H3K27me3 is observed in both the fragments from the old vegetative MACs and the developing MACs.

Although I agree that Ezl1 mediates H3K27me3 and H3K9me3 (directly or indirectly) in *P. tetraurelia* based on the *in vivo* data, there are some major issues to be addressed as detailed above. I would not recommend this manuscript for publication.

Reviewers' comments:

Reviewer #2 (Remarks to the Author):

I am glad that this time the authors have taken the comments seriously and have performed many additional experiments and analysis to address my concerns. This manuscript has been significantly improved and I support its publication.

Reviewer #3 (Remarks to the Author):

Having looked through the revised m/s my impression is that the m/s has been considerably improved and the criticisms that were raised have been well addressed. In vitro assays have been done using *Paramecium* H3 as well as *Xenopus* H3 and the K27R mutations do not substantially reduce H3 SAM incorporation, which suggests that this is not the only methylation catalysed. The K27R K9R double mutant causes some drop in radioactivity incorporation, but by no means eliminates it, suggesting other residues are modified, or conceivably there is a contaminating activity. Silver stained gels of tandem affinity purified complex suggest reasonable purity. It would be nice to know what the other bands correspond to, but this will likely be the subject of another paper and that seems fair enough, assuming that none of the other components correspond to SET domain proteins. The mass spec comparison of wt and mutant Ezh1 complexes I think adds a lot, giving more confidence that the complex can methylate H3K9 and that this is much reduced in the mutant form of the complex, so unlikely to be due to a contaminant protein. It also suggests that other lysine residues can be methylated, which may explain why the H3K9R K27R mutations do not eliminate H3 SAM incorporation. The RNA seq and ChIP seq data strengthens the case that H3K9me3 and H3K27me3 can co occur at TE and show reduced levels/increased expression in Ezh1 mutants. Overall I find it an interesting paper and think it is very suitable for publication in *Nature Comm*

Reviewer #5 (Remarks to the Author):

The manuscript "A Polycomb protein mediates dual methylation of H3 lysines 9 and 27 to ensure transposable element repression" by Frapporti et al. characterizes the Enhancer-of-zeste-like protein Ezh1 from the unicellular eukaryote *Paramecium tetraurelia*. This protein is a structural homolog of the human EZH2 protein, the histone H3K27 methyltransferase. The authors show that Ezh1 can catalyze the trimethylation of histone H3K9 and H3K27 in vitro (to a minor extent H3K23 as well). The authors further show that both H3K9me3 and H3K27me3 co-occur at transposable elements in an Ezh1-dependent manner, causing re-expression of these genes upon Ezh1 loss. Overall, the manuscript is well written and experiments completed in a clear manner. While it is clear that Ezh1 activity affects both H3K27 and H3K9 methylation patterns (in vitro and in vivo), the data has not quite convinced me that Ezh1 is a strong H3K9 methyltransferase, and that there is not another methyltransferase in the purified complex. Until the authors identify all the components in the purified complex, I cannot recommend publication as the authors have not ruled out that H3K9 methylation is indirect. My comments are as noted below.

1. I am little concerned that the authors state that even in the presence of a double K9A/K27A mutant, there is still 15% residual activity. The authors show H3K23 methyl could be methylated in the double mutant. This makes me feel as though the enzyme complex is not pure, or else the Ezh1 methyltransferase is very promiscuous. Have the authors identified all of the components from the Flag-HA-Ezh1 pulldowns? Could there be another methyltransferase co-purifying with Ezh1? What is the composition of the Ezh1 complex?
2. The in vitro *Xenopus* versus *Paramecium* data is odd. Seems as though Ezh2 is more sensitive to the double K9/K27 mutant in *Xenopus* than *Paramecium*, why? There is still a lot of activity with the double mutant in *Paramecium*. Is this all H3K23 or other residues?
3. In the immunofluorescence images, it seems as though in vivo there is very little H3K9me3 compared to H3K27me3, while in the in vitro assays it's closer to being even? What are the natural

abundances of these marks in vivo?

4. Depletion of Ezh1 with RNAi does decrease the presence of H3K9me3 and H3K27me3 at transposable elements, but its still not clear if this is direct or indirect with respects to both marks. I am not convinced that H3K9me3 is not indirect.

We are very satisfied to learn that the previous reviewers of our article, whom we thank for pushing us to do the additional experiments, accept the present version. We address the concerns raised by reviewer #5 in what follows. All changes are highlighted in the manuscript text file.

We have also modified Figures 2-3-5 and Supplementary Figure 2 to follow Nature Communications policy and represent the data with individual data points whenever possible. In addition, we provide raw data for all western blots and gels (Figure 2 and Supplementary Fig. 2), and source data underlying Figures 2-3-4-5 and Supplementary Figure 2, as a Source Data file.

Response to Reviewer #5:

1. I am little concerned that the authors state that even in the presence of a double K9A/K27A mutant, there is still 15% residual activity. The authors show H3K23 methyl could be methylated in the double mutant. This makes me feel as those the enzyme complex is not pure, or else the Ezh1 methyltransferase is very promiscuous. Have the authors identified all of the components from the Flag-HA-Ezh1 pulldowns? Could there be another methyltransferase co-purifying with Ezh1? What is the composition of the Ezh1 complex?

We agree with the reviewer that the 15% residual activity detected with the double K9A/K27A *Xenopus* H3 mutant (Fig. 2 panel C) may represent a contaminating activity in our GFP-pulldown preparations or, alternatively, a weak activity of Ezh1 toward other residues of H3, as discussed in the Results section “Ezh1 is a methyltransferase that targets both lysine 9 and lysine 27 of histone H3” line 197. To distinguish between these possibilities, we purified Ezh1 from soluble nuclear protein extracts using very stringent immunopurification methodologies. This indicates that our preparations are highly pure and consistent between replicates.

Furthermore:

- i) *In vitro* incorporation of radioactive SAM dropped with the Ezh1^{H526A} mutant, which carries a point mutation within the catalytic SET domain (Fig. 2; Supplementary Fig. 2). This observation makes it unlikely that another methyltransferase contaminates our preparations.
- ii) *In vitro* assays done with the double-affinity purified Ezh1 complexes confirmed the results we had previously obtained with GFP-Trap purification. The HMT activity towards H3 is greatly diminished with the *Xenopus* H3 K9A/K27A double mutant (Supplementary Fig. 2 panel D), but not totally abolished. We believe that it is unlikely that the same contaminant activity would be pulled down through these different purification strategies.
- iii) Using *Paramecium* histone H3 as a substrate, ³H-SAM incorporation dropped with the K9R/K27R double-mutant H3 (Figure 2 panels D and E; and Supplementary Figure 2 panel D), but did not eliminate it, suggesting that other residues can be modified regardless of the substrates.
- iv) Mass spectrometry allowed us to determine the sites of methylation using the double-affinity purified Ezh1 complexes: both K9 and K27 of *Paramecium* H3 are methylated in an Ezh1-dependent manner. Methylation of K9 is strongly reduced in the mutant Ezh1, making it unlikely to be due to a contaminant protein.
- v) Mass spectrometry analysis revealed that the residual methylation detected using the double K9R/K27R *Paramecium* H3 mutant can be attributed to a weak activity of Ezh1 toward K23 (Fig. 2 panel E), an effect that is much more pronounced when positions 9 and 27 are mutated and can no longer be methylated (Supplementary Table 1).

Altogether, these data strongly support the hypothesis that Ezh1 has broader substrate specificity than Ezh2. This is stated in the Discussion, end of Section “H3K27 and H3K9 specificity of an Enhancer-of-zeste-like protein” line 462.

To make it clearer, we have rephrased the last paragraph of the Introduction (line 79). It now reads:

“Here, we provide strong evidence that, Paramecium Ezl1, which exhibits significant sequence and structural similarities with human Ezh2, displays distinct enzymatic properties. The Ezl1 methyltransferase has a broader substrate specificity, including histone H3 K27 and K9, and potentially secondary residues in vitro.”

Also, we have now analyzed by mass spectrometry the composition of the Ezl1 complex from our Flag-HA-Ezl1 pull down experiments. Silver stained gels of tandem affinity purified complex indicate a high degree of purity. We confirmed that Ezl1 is the only SET-domain containing protein (thoroughly annotated in a previous study, *Lhuillier-Akakpo et al.* 2014) that could be identified in our preparations. Some of the other proteins identified to interact with Ezl1 could be compatible with the formation of a “PRC2” type of complex. However, due to large sequence divergence, further investigations are required to report the exact composition of the complex. This will be the subject of a new grant and is beyond the scope of this publication.

2. The in vitro *Xenopus* versus *Paramecium* data is odd. Seems as though Ezh2 is more sensitive to the double K9/K27 mutant in *Xenopus* than *Paramecium*, why? There is still a lot of activity with the double mutant in *Paramecium*. Is this all H3K23 or other residues?

As noted by the reviewer, the incorporation of radioactive SAM drops with the double H3 mutant compared to wild type or single mutant, but more residual activity appears detectable with the *Paramecium* double mutant H3 than with the *Xenopus* double mutant H3 (Supplementary Fig. 2 panel D). Yet, it is important to note that the comparison is limited by the fact that the mutants are not exactly the same (K9A/K27A for *Xenopus* and K9R/K27R for *Paramecium*). Whether differences in the N-terminal amino acid sequence between the two proteins (Supplementary Fig. 2 panel E) could also contribute to this result remains to be investigated.

To acknowledge the difference observed between the *Paramecium* double mutant H3 and the *Xenopus* double mutant H3, we have added the following sentences:

“More residual activity appears detectable with the Paramecium double mutant H3 (lane 9) than with the Xenopus double mutant H3 (lane 5). This difference may be due to the fact that the mutants are not exactly the same (K9A/K27A for Xenopus and K9R/K27R for Paramecium) or to differences in the N-terminal amino acid sequence between the two proteins (see panel e), or both.”

We decided to include this comment in the legend of Supplementary Fig. 2 panel D, where the data is shown. We felt it would otherwise interrupt the flow of the main text.

Nonetheless, other lysine residues within the N-terminal part of histone H3, besides lysine 9 and lysine 27, can be methylated *in vitro*. Indeed, our mass spectrometry experiments establish that K23 is methylated in an Ezl1-dependent manner when lysines 9 and 27 are no longer substrates for methylation by the enzyme (Supplementary Table 1). This is mentioned in the Results, end of section “Ezl1 is a methyltransferase that targets both lysine 9 and lysine 27 of histone H3” line 232.

To make this clearer, we have rephrased the last paragraph of the Results section “Ezl1 is a methyltransferase that targets both lysine 9 and lysine 27 of histone H3” line 234. It now reads:

“Altogether, our data support the hypothesis that lysines 9 and 27 are both major substrates of Ezl1, while K23 is a secondary site of methylation whose in vivo relevance would require further investigation.”

3. In the immunofluorescence images, it seems as though in vivo there is very little H3K9me3 compared to H3K27me3, while in the in vitro assays its closer to being even? What are the natural abundances of these marks in vivo?

The reviewer may not be familiar with the unconventional organism used in our study, which is characterized by the coexistence of different types of nuclei in the same cytoplasm. After sexual events, when DNA elimination occurs, three types of nuclei co-exist: the fragmented maternal somatic MAC, the new developing MACs and the new MICs (as depicted on the scheme Figure 3). The reviewer is right in noting that H3K9me3 and H3K27me3 have a distinct distribution in the cell. Immunostaining experiments showed that H3K9me3 is exclusively detected in the two new developing MACs, while H3K27me3 is detected in the maternal MAC and in the two new developing MACs (Fig. 3). This is explained in the legend of Fig. 3. A more detailed description of their localization throughout the life cycle can be found in *Lhuillier-Akakpo et al.* 2014.

Yet, quantitative differences cannot be evaluated due to the use of different antibodies against H3K27me3 and H3K9me3. To our knowledge, the only way to get a thorough evaluation of the abundance of these marks would be to perform mass spectrometry in the presence of peptides mimicking the different modifications of K9 and K27. This approach is not straightforward in *Paramecium* owing to the presence of several histone variants (not yet characterized) and consequent uncertainty regarding the peptides to include.

We have changed the wording in the Results (line 253) to remind the reader that different antibodies were used for H3K9me3 and H3K27me3 immunofluorescence experiments, precluding quantitative comparison of abundance of these marks. It now reads:

“During development, H3K9me3 and H3K27me3 deposition were assayed by immunostaining with anti-H3K9me3 and anti-H3K27me3 antibodies, respectively.”

Finally, although whether H3K9me3 and H3K27me3 have different abundances *in vivo* is an interesting question, it is not necessary to address this question to conclude that i) the Ezl1 enzyme is capable of catalyzing the deposition of both marks *in vitro* in a comparable manner and ii) the appearance of both marks during normal development depends on the catalytic activity of Ezl1.

4. Depletion of Ezl1 with RNAi does decrease the presence of H3K9me3 and H3K27me3 at transposable elements, but it is still not clear if this is direct or indirect with respects to both marks. I am not convinced that H3K9me3 is not indirect.

We acknowledge that the depletion of Ezl1 *in vivo* does not formally demonstrate that Ezl1 catalyzes H3K27me3 and H3K9me3 directly. Nonetheless, we think that the most parsimonious explanation, given the body of experimental evidence obtained in this study, is that H3K9 and H3K27 are both recognized as substrates by the *Paramecium* Ezl1 protein. Indeed, we observed that:

- i) Ezl1 purified from *Paramecium* nuclear extracts catalyzes *in vitro* methylation of *Xenopus* and *Paramecium* histone H3 (Fig. 2; Supplementary Fig. 2).
- ii) Ezl1 *in vitro* methyltransferase activity is reduced using K9/27double mutant H3 compared to wild-type H3 or single mutants (Fig. 2; Supplementary Fig. 2).
- iii) This *in vitro* activity is severely compromised in the Ezl1^{H526A} mutant, which carries a substitution within the catalytic SET domain (Fig. 2; Supplementary Fig. 2).
- iv) *In vivo* deposition of H3K9me3 and H3K27me3 are no longer detected in the Ezl1^{H526A} mutant (Fig. 3).

We edited the Discussion (line 431) as follows:

*“Although we cannot formally exclude the possibility that the effect of *in vivo* Ezl1 depletion on the methylation of a specific lysine is indirect, we believe that the most parsimonious explanation, given the body of experimental evidence obtained in this study, is that H3K9 and H3K27 are both recognized as substrates by the *Paramecium* Ezl1 protein.”*

In addition to the changes indicated above, we also added for clarity the following sentence in the Results section (line 173):

*“Purified PRC2 is known to methylate histone H3 on K27 and K9 *in vitro*, even though the activity towards K9 is low⁷⁻⁹.”*

REVIEWERS' COMMENTS:

Reviewer #5 (Remarks to the Author):

The author has satisfied all of my previous concerns with this revised manuscript. I now recommend this manuscript for publication.